# A Feasible Level Proximal Point Method for Nonconvex Sparse Constrained Optimization

**Digvijay Boob***
Southern Methodist University
Dallas, TX
dboob@smu.edu

**Qi Deng**
Shanghai university of Finance & Economics
Shanghai, China
qideng@sufe.edu.cn

**Guanghui Lan**
Georgia Tech
Atlanta, GA
george.lan@isye.gatech.edu

**Yilin Wang**
Shanghai university of Finance & Economics
Shanghai, China
2017110765@live.sufe.edu.cn

## Abstract

Nonconvex sparse models have received significant attention in high-dimensional machine learning. In this paper, we study a new model consisting of a general convex or nonconvex objectives and a variety of continuous nonconvex sparsity-inducing constraints. For this constrained model, we propose a novel proximal point algorithm that solves a sequence of convex subproblems with gradually relaxed constraint levels. Each subproblem, having a proximal point objective and a convex surrogate constraint, can be efficiently solved based on a fast routine for projection onto the surrogate constraint. We establish the asymptotic convergence of the proposed algorithm to the Karush-Kuhn-Tucker (KKT) solutions. We also establish new convergence complexities to achieve an approximate KKT solution when the objective can be smooth/nonsmooth, deterministic/stochastic and convex/nonconvex with complexity that is on a par with gradient descent for unconstrained optimization problems in respective cases. To the best of our knowledge, this is the first study of the first-order methods with complexity guarantee for nonconvex sparse-constrained problems. We perform numerical experiments to demonstrate the effectiveness of our new model and efficiency of the proposed algorithm for large scale problems.

## 1 Introduction

Recent years have witnessed a great deal of work on the sparse optimization arising from machine learning, statistics and signal processing. A fundamental challenge in this area lies in finding the best set of size $k$ out of a total of $d$ $(k < d)$ features to form a parsimonious fit to the data:

$$\min \ \psi(x), \quad \text{subject to} \quad \|x\|_0 \leqslant k, x \in \mathbb{R}^d. \tag{1}$$

However, due to the discontinuity of $\|\cdot\|_0$ norm[2], the above problem is intractable when there is no other assumptions. To bypass this difficulty, a popular approach is to replace the $\ell_0$-norm by the $\ell_1$-norm, giving rise to an $\ell_1$-constrained or $\ell_1$-regularized problem. A notable example is the Lasso

([31]) approach for linear regression and its regularized variant

$$\min \|b - Ax\|_2^2, \qquad \text{subject to } \|x\|_1 \leqslant \tau, x \in \mathbb{R}^d; \tag{2}$$

$$\min \|b - Ax\|_2^2 + \lambda \|x\|_1. \tag{3}$$

Due to the Lagrange duality theory, problem (2) and (3) are equivalent in the sense that there is a one-to-one mapping between the parameters $\tau$ and $\lambda$. A substantial amount of literature already exists for understanding the statistical properties of $\ell_1$ models ([41, 32, 7, 39, 19]) as well as for the development efficient algorithms when such models are employed ([11, 1, 22, 34, 19]).

In spite of their success, $\ell_1$ models suffer from the issue of biased estimation of large coefficients [12] and empirical merits of using nonconvex approximations were shown in [26]. Due to these observations, a large body of recent research looked at replacing the $\ell_1$-penalty in (3) by a nonconvex function $g(x)$ to obtain sharper approximation of the $\ell_0$-norm:

$$\min \ \psi(x) + \beta g(x), \tag{4}$$

where , throughout the paper, $g(x)$ is a nonsmooth nonconvex function of the form

$$g(x) = \lambda \|x\|_1 - h(x).$$

Here $h(x)$ is a convex and continuously differentiable function, giving $g(x)$ a DC form. This class of constraints already covers many important nonconvex sparsity inducing functions in the literature (see Table 2).

Despite the favorable statistical properties ([12, 38, 8, 40]), nonconvex models have posed a great challenge for optimization algorithms and has been increasingly an important issue ([36, 16, 17, 29]). While most of these works studied the regularized version, it is often favorable to consider the following constrained form:

$$\min \ \psi(x), \quad \text{subject to} \quad g(x) \leqslant \eta, x \in \mathbb{R}^d, \tag{5}$$

since sparsity of solutions is imperative in many applications of statistical learning and constrained form in (5) explicitly imposes such a requirement. In contrast, (4) imposes sparsity implicitly using penalty parameter $\beta$. However, unlike the convex problems, large values of $\beta$ do not necessarily imply small value of the nonconvex penalty $g(x)$.

Therefore, it is natural to ask *whether we can provide an efficient algorithm for problem* (5). The continuous nonconvex relaxation (5) of the $\ell_0$-norm in (1), albeit a straightforward one, was not studied in the literature. We suspect that to be the case due to the difficulty in handling nonconvex constraints algorithmically. There are two theoretical challenges: First, since the regularized form (4) and the constrained form (5) are not equivalent due to the nonconvexity of $g(x)$, we cannot bypass (5) by solving problem (4) instead. Second, the nonconvex function $g(x)$ can be nonsmooth especially for the sparsity applications, presenting a substantial challenge for classic nonlinear programming methods, e.g., augmented Lagrangian methods and penalty methods (see [2]) which assumes that functions are continuously differentiable.

**Our contributions**    In this paper, we study the newly proposed nonconvex constrained model (5). In particular, we present a novel level-constrained proximal point (LCPP) method for problem (5) where the objective $\psi$ can be either deterministic/stochastic, smooth/nonsmooth and convex/nonconvex and the constraint $\{g(x) \leqslant \eta\}$ models a variety of sparsity inducing nonconvex constraints proposed in the literature. The key idea is to translate problem (5) into a sequence of convex subproblems where $\psi(x)$ is *convexified* using a proximal point quadratic term and $g(x)$ is *majorized* by a convex function $\widetilde{g}(x)[\geqslant g(x)]$. Note that $\{\widetilde{g}(x) \leqslant \eta\}$ is a convex subset of the nonconvex set $\{g(x) \leqslant \eta\}$.

We show that starting from a strict feasible point[3], LCPP traces a feasible solution path with respect to the set $\{g(x) \leqslant \eta\}$. We also show that LCPP generates convex subproblems for which bounds on the optimal Lagrange multiplier (or the optimal dual) can be provided under a mild and a well-known constraint qualification. This bound on the dual and the proximal point update in the objective allows us to prove asymptotic convergence to the KKT points of the problem (5).

While deriving the complexity, we consider the inexact LCPP method that solves convex subproblems approximately. We show that the constraint, $\widetilde{g}(x) \leqslant \eta$, has an efficient projection algorithm.

Table 1: Iteration complexities of LCPP for problem (5) when the objective can be either convex or nonconvex, smooth or nonsmooth and deterministic or stochastic

| | Convex (5) | | Nonconvex (5) | |
|---|---|---|---|---|
| Cases | Smooth | Nonsmooth | Smooth | Nonsmooth |
| Deterministic | $O(1/\varepsilon)$ | $O(1/\varepsilon^2)$ | $O(1/\varepsilon)$ | $O(1/\varepsilon^2)$ |
| Stochastic | $O(1/\varepsilon^2)$ | $O(1/\varepsilon^2)$ | $O(1/\varepsilon^2)$ | $O(1/\varepsilon^2)$ |

Hence, each convex subproblem can be solved by projection-based first-order methods. This allows us to be feasible even when the solution reaches arbitrarily close to the boundary of the set $\{g(x) \leqslant \eta\}$ which entails that the bound on the dual mentioned earlier works in the inexact case too. Moreover, efficient projection-based first-order method for solving the subproblem helps us get an accelerated convergence complexity of $O(\frac{1}{\varepsilon})[O(\frac{1}{\varepsilon^2})]$ gradient [stochastic gradient] in order to obtain an $\varepsilon$-KKT point. In particular, refer to Table 1. We see that in the case where objective is smooth and deterministic, we obtain convergence rate of $O(1/\varepsilon)$ whereas for nonsmooth and/or stochastic objective we obtain convergence rate of $O(1/\varepsilon^2)$. This complexity is nearly the same as that of the gradient [stochastic gradient] descent for the regularized problem (4) of the respective type. Remarkably, this convergence rate is better than black-box nonconvex function constrained optimization methods proposed in the literature recently ([5, 21]). See related work section for more detailed discussion. Note that the convergence of gradient descent does not ensure a bound on the infeasibility of the constraint $g$, whereas the KKT criterion requires feasibility on top of stationarity. Moreover, such a bound cannot be ensured theoretically due to the absence of duality. Hence, our algorithm provides additional guarantees without paying much in the complexity.

We perform numerical experiments to measure the efficiency of our LCPP method and the effectiveness of the new constrained model (5). First, we show that our algorithm has competitive running time performance against open-source solvers, e.g., DCCP [27]. Second, we also compare the effectiveness of our constrained model with respect to the existing convex and nonconvex regularization models in the literature. Our numerical experiments show promising results compared to $\ell_1$-regularization model 3 and has competitive performance with respect to recently developed algorithm for nonconvex regularization model 4 (see [16]). Given that this is the first study in the development of algorithms for the constrained model, we believe empirical study of even more efficient algorithms solving problem (5) may be of independent interest and can be pursued in the future.

**Related work**    There is a growing interest in using convex majorization for solving nonconvex optimization with nonconvex function constraints. Typical frameworks include difference-of-convex (DC) programming ([30]), majorization-minimization ([28]) to name a few. Considering the substantial literature, we emphasize the most relevant work to our current paper. Scutari et al. [26] proposed general approaches to majorize nonconvex constrained problems and include (5) as a special case. They require exact solutions of the subproblems and prove asymptotic convergence which is prohibitive for large-scale optimization. Shen et al. [27] proposed a disciplined convex-concave programming (DCCP) framework for a class of DC programs in which (5) is a special case. Their work is empirical and does not provide specific convergence results.

The more recent works [5, 21] considered a more general problems where $g(x) = \widetilde{h}(x) - h(x)$ for some general convex function $\widetilde{h}$. They propose a type of proximal point method in which large enough quadratic proximal term is added into both objective and constraint in order to obtain a convex subproblem. This convex function constrained subproblem can be solved by oracles whose output solution might have small infeasibility. Moreover these oracles have weaker convergence rates due to generality of function $\widetilde{h}$ over $\ell_1$. Complexity results proposed in these works, when applied to problem (5), entail $O(1/\varepsilon^{3/2})$ iterations for obtaining an $\varepsilon$-KKT point under a *strong feasibility* constraint qualification. In similar setting, we show faster convergence result of $O(1/\varepsilon)$. This due to the fact that our oracle for solving the subproblem is more efficient than those used in their paper. We can obtain such an oracle due to two reasons: i) convex surrogate constraint $\widetilde{g}$ in LCPP majorizes the constraint differently than adding the proximal quadratic term, ii) presence of $\ell_1$ in the form of $g(x)$ allows for developing an efficient projection mechanism onto the chosen form of $\widetilde{g}$. Moreover, our convergence results hold under a well-known constraint qualification which is weaker compared to

Figure 1: Graphs for various constraints along with $\ell_1$. For $\ell_p(0 < p < 1)$, we have $\varepsilon = 0.1$ .

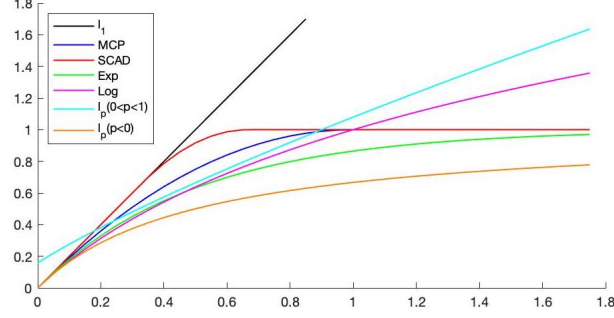

*strong feasibility* since our oracle outputs a feasible solution whereas they can get a solution which is slightly infeasible.

There is also a large body of work on directly optimizing the $\ell_0$ constraint problem [3, 4, 13, 37, 42]. While [3] can be quite good for small dimension $d = 1000$s, it remains unclear how to scale up for larger datasets. Other methods are part of the hard-thresholding algorithms, requiring additional assumptions such as *Restricted Isometry Property*. These research areas, though interesting, are not related to the continuous optimization setting where large-scale problems can be solved relatively easily. Henceforth, we only focus on the continuous approximations of $\ell_0$-norm.

**Structure of the paper**     Section 2 presents the problem setup and preliminaries. Section 3 introduces LCPP method and shows the asymptotic convergence, convergence rates and the boundedness of the optimal dual. Section 4 presents numerical results. Finally, Section 5 draws conclusion.

## 2   Problem setup

Our main goal is to solve problem (5). We make Assumption 2.1 throughout the paper.

**Assumption 2.1.**  *1. $\psi(x)$ is a continuous and possibly nonsmooth nonconvex function satisfying:*

$$\psi(x) \geqslant \psi(y) + \left\langle \psi'(y), x - y \right\rangle - \tfrac{\mu}{2} \left\| x - y \right\|^2 . \tag{6}$$

*2. $g(x)$ is a nonsmooth nonconvex function of the form $g(x) = \lambda\|x\|_1 - h(x)$, where $h(x)$ is convex and continuously differentiable.*

Table 2: Examples of constraint function $g(x) = \lambda\|x\|_1 - h(x)$.

| Function $g(x)$ | Parameter $\lambda$ | Function $h(x)$ |
|---|---|---|
| MCP[38] | $\lambda$ | $h_{\lambda,\theta}(x) = \begin{cases} \frac{x^2}{2\theta} & \text{if } |x| \leqslant \theta\lambda, \\ \lambda\,|x| - \frac{\theta\lambda^2}{2} & \text{if } |x| > \theta\lambda. \end{cases}$ |
| SCAD[12] | $\lambda$ | $h_{\lambda,\theta}(x) = \begin{cases} 0 & \text{if } |x| \leqslant \lambda, \\ \frac{x^2 - 2\lambda|x| + \lambda^2}{2(\theta-1)} & \text{if } \lambda < |x| \leqslant \theta\lambda, \\ \lambda|x| - \frac{1}{2}(\theta+1)\lambda^2 & \text{if } |x| > \theta\lambda. \end{cases}$ |
| Exp[6] | $\lambda$ | $h_\lambda(x) = e^{-\lambda|x|} - 1 + \lambda|x|.$ |
| Log[33] | $\frac{\theta}{\log(1+\theta)}$ | $h_\theta(x) = \frac{\theta}{\log(1+\theta)}|x| - \frac{\log(1+\theta|x|)}{\log(1+\theta)}.$ |
| $\ell_p(0<p<1)$[14] | $\frac{\varepsilon^{1/\theta-1}}{\theta}$ | $h_{\varepsilon,\theta}(x) = \frac{\varepsilon^{1/\theta-1}}{\theta}|x| - (|x|+\varepsilon)^{1/\theta}.$ |
| $\ell_p(p<0)$[24] | $-p\theta$ | $h_\theta(x) = -p\theta|x| - 1 + (1+\theta|x|)^p.$ |

**Notations**     We use $\|\cdot\|$ to denote standard Euclidean norm whereas $\ell_1$-norm is denoted as $\|\cdot\|_1$. The Lagrangian function for problem (5) is defined as $\mathcal{L}(x, y) = \psi(x) + y(g(x) - \eta)$ where

$y \geqslant 0$. For nonconvex nonsmooth function $g(x)$ in the form of (2), we denote its *subdifferential*[4] by $\partial g(x) = \partial(\lambda \|x\|_1) - \nabla h(x)$. For this definition of subdifferential, we consider the following KKT condition:

**The KKT condition**   For Problem (5), we say that $x$ is the (stochastic) $(\varepsilon, \delta)$- KKT solution if there exists $\bar{x}$ and $\bar{y} \geqslant 0$ such that $g(\bar{x}) \leqslant \eta$, $\mathbb{E} \|x - \bar{x}\|^2 \leqslant \delta$

$$\mathbb{E} |\bar{y} [g(\bar{x}) - \eta]| \leqslant \varepsilon$$
$$\mathbb{E} [\text{dist} (\partial_x \mathcal{L}(\bar{x}, \bar{y}), 0)]^2 \leqslant \varepsilon \tag{7}$$

Moreover, for $\varepsilon = \delta = 0$, we have that $\bar{x}$ is the KKT solution or satisfied KKT condition. If $\delta = O(\varepsilon)$, we refer to this solution as an $\varepsilon$-KKT solution in order to be brief.

It should be mentioned that local or global optimality does not generally imply the KKT condition. However, KKT condition is shown to be necessary for optimality when Mangasarian-Fromovitz constraint qualification (MFCQ) holds [5]. Below, we make MFCQ assumption precise:

**Assumption 2.2** (MFCQ [5]). *Whenever the constraint is active: $g(\bar{x}) = \eta$, there exists a direction $z$ such that $\max_{v \in \partial g(\bar{x})} v^T z < 0$.*

For differentiable $g$, MFCQ requires existence of $z$ such that $z^T \nabla g(\bar{x}) < 0$, reducing to the classical form of MFCQ [2]. Below, we summarize necessary optimality condition under MFCQ.

**Proposition 2.3** (Necessary condition [5]). *Let $\bar{x}$ be a local optimal solution of problem* (5). *If $\bar{x}$ satisfies Assumption 2.2, then there exists $\bar{y} \geqslant 0$ such that* (7) *holds with $\varepsilon = \delta = 0$.*

## 3   A novel proximal point algorithm

---
**Algorithm 1** Level constrained proximal point (**LCPP**) method
---
1: **Input:** $x^0 = \hat{x}$, $\gamma > 0$, $\eta_0 < \eta$
2: **for** $k = 1$ **to** $K$ **do**
3:     Set $\eta_k = \eta_{k-1} + \delta_k$;
4:     $g_k(x) := \lambda \|x\|_1 - h(x^{k-1}) - \nabla h(x^{k-1})^T (x - x^{k-1})$;
5:     Return feasible solution $x^k$ of the problem
$$\min \psi_k(x) = \psi(x) + \tfrac{\gamma}{2} \|x - x^{k-1}\|^2, \qquad \text{subject to} \quad g_k(x) \leqslant \eta_k \tag{8}$$
6: **end for**
---

LCPP method solves a sequence of convex subproblems (8). In particular, note that $g_k(x)$ majorizes $g(x)$: $g_k(x) \geqslant g(x)$, $g_k(x^{k-1}) = g(x^{k-1})$. implying that $\{g_k(x) \leqslant \eta_k\}$ is a convex subset of the original problem. It can also be observed that adding a proximal term in the objective yields $\psi_k$ strongly convex for large enough $\gamma > 0$. In the current form, Algorithm 1 requires a feasible solution of (8) and requirement of sequence $\{\eta_k\}$ is left unspecified. We first make the following assumptions.

**Assumption 3.1** (Strict feasibility). *There exist sequence $\{\eta_k\}_{k \geqslant 0}$ satisfying:*
*1. $\eta_0 < \eta$ and a point $\hat{x}$ of such that $g(\hat{x}) < \eta_0$.*
*2. The sequence $\{\eta_k\}$ is monotonically increasing and converges to $\eta$: $\lim_{k \to \infty} \eta_k = \eta$.*

In light of Assumption 3.1, starting from a strictly feasible point $x^0$, Algorithm 1 solves subproblems (8) with gradually relaxed constraint levels. This allows us to assert that each subproblem is strictly feasible[5]. Indeed, we have $g_k(x^k) \leqslant \eta_k \Rightarrow g_{k+1}(x^k) = g(x^k) \leqslant g_k(x^k) \leqslant \eta_k < \eta_{k+1}$. This implies the existence of KKT solution for each subproblem. A formal statement can be found in the appendix. Moreover, all the proofs of our technical results can be found in the appendix and we just make statements in the main article henceforth.

**Asymptotic convergence of LCPP method and boundedness of the optimal dual**

Our next goal is to establish asymptotic convergence of Algorithm 1 to the KKT points. To this end, we require a uniform boundedness assumption on the Lagrange multipliers. First, we prove asymptotic convergence under this assumption then we justify it under MFCQ. Before stating the convergence results, we make the following boundedness assumption.

**Assumption 3.2** (Boundedness of dual variables). *There exists $B > 0$ such that $\sup_k \bar{y}^k < B$ a.s.*

For the deterministic case, we remove the measurablity part in the above assumption and assert that $\sup_k \bar{y}^k < B$. The following asymptotic convergence theorem is in order.

**Theorem 3.3** (Convergence to KKT). *Let $\pi_k$ denotes the randomness of $x^1, x^2, ..., x^{k-1}$. Assume that there exists a $\rho \in [0, \gamma - \mu]$ and a summable nonnegative sequence $\zeta_k$ such that*

$$\mathbb{E}[\psi_k(x^k) - \psi_k(\bar{x}^k)|\pi_k] \leqslant \tfrac{\rho}{2}\|\bar{x}^k - x^{k-1}\|^2 + \zeta_k. \tag{9}$$

*Then, under Assumption 3.1 and 3.2 for any limit point $\widetilde{x}$ of the proposed algorithm, there exists a dual variable $\widetilde{y}$ such that $(\widetilde{x}, \widetilde{y})$ satisfies KKT condition, almost surely.*

This theorem shows that any limit point of Algorithm 1 converges to a KKT point. However, it makes the assumption that dual is bounded. Since the optimal dual depends on the convex subproblems (8) which are generated dynamically in the algorithm, it is important to justify Assumption 3.2. To this end, we show that Assumption 3.2 is satisfied under a well-known constraint qualification.

**Theorem 3.4** (Boundedness condition). *Suppose Assumption (3.1) and relation (9) are satisfied and all limit points of Algorithm 1 exists a.s., and satisfy the MFCQ condition. Then, $\bar{y}^k$ is bounded a.s.*

This theorem shows the existence of dual under the MFCQ assumption for all limit points of Algorithm 1. MFCQ is a mild constraint qualification frequently used in the existing literature [2]. In certain cases, we also provide explicit bounds on the dual variables using the fact that origin is most feasible solution to the subproblem. These bounds quantify how "closely" the MFCQ assumption is violated and provides explicitly the effect on the magnitude of the optimal dual. Additional results and discussion in this regard are deferred to the Appendix B. For the purpose of this article, we assume that the dual variables remain bounded henceforth.

**Complexity of LCPP method**

Our goal here is to analyze the complexity of the proposed algorithm. Apart from the negative lower curvature guarantee (6) of the objective function, we impose that $h$ has Lipschitz continuous gradients, $\|\nabla h(x) - \nabla h(y)\| \leqslant L_h \|x - y\|$. This is satisfied by all functions in Table 2. Now we discuss a general convergence result of LCPP method for original nonconvex problem (5).

**Theorem 3.5.** *Suppose Assumption 3.1 and 3.2 hold such that $\delta_k = \frac{\eta - \eta_0}{k(k+1)}$ for all $k \geqslant 1$. Let $x^k$ satisfy (9) where $\rho \in [0, \gamma - \mu)$ and $\{\zeta_k\}$ is a summable nonnegative sequence. Moreover, $x^k$ is a feasible solution of the $k$-th subproblem, i.e.,*

$$g_k(x^k) \leqslant \eta_k. \tag{10}$$

*If $\hat{k}$ is chosen uniformly at random from $\left\lfloor \frac{K+1}{2} \right\rfloor$ to $K$ then there exists a pair $(\bar{x}^{\hat{k}}, \bar{y}^{\hat{k}})$ satisfying*

$$\mathbb{E}[\text{dist}(\partial_x \mathcal{L}(\bar{x}^{\hat{k}}, \bar{y}^{\hat{k}}), 0)^2] \leqslant \tfrac{16(\gamma^2 + B^2 L_h^2)}{K(\gamma - \mu - \rho)} \left( \tfrac{\gamma - \mu + \rho}{2(\gamma - \mu)} \Delta^0 + Z \right),$$

$$\mathbb{E}[\bar{y}^{\hat{k}}|g(\bar{x}^{\hat{k}}) - \eta|] \leqslant \tfrac{2BL_h}{K(\gamma - \mu - \rho)} \left( \tfrac{\gamma - \mu + \rho}{\gamma - \mu} \Delta^0 + 2Z \right) + \tfrac{2B(\eta - \eta_0)}{K},$$

$$\mathbb{E}[\|x^{\hat{k}} - \bar{x}^{\hat{k}}\|^2] \leqslant \tfrac{4\rho(\gamma - \mu + \rho)}{K(\gamma - \mu)^2(\gamma - \mu - \rho)} \Delta^0 + \tfrac{8Z}{K(\gamma - \mu - \rho)},$$

*where, $\Delta^0 := \psi(x^0) - \psi(x^*)$, $Z := \sum_{k=1}^{K} \zeta_k$ and expectation is taken over the randomness of $\hat{k}$ and solutions $x^k$, $k = 1, \dots, K$.*

Note that Theorem 3.5 assumes that subproblem (8) can be solved according to the framework of (9) and (10). When the subproblem solver is deterministic then we ignore the expectation in (9). It is easy to see from the above theorem that for $x^{\hat{k}}$ to be an $\varepsilon$-KKT point, we must have $K = O(1/\varepsilon)$ and $\zeta_k$ must be small enough such that $Z$ is bounded above by a constant. The complexity analysis of different cases now boils down to understanding the number of iterations of the subproblem solver needed in order to satisfy these requirements on $\rho$ and $\{\zeta_k\}$ (or $Z$).

In the rest of this section, we provide a unified complexity result for solving subproblem (8) in Algorithm 1 such that criteria in (9) and (10) are satisfied for various settings of the objective $\psi(x)$.

**Unified method for solving subproblem** (8)    Here we provide a unified complexity analysis for solving subproblem (8). In particular, consider the form of the objective $\psi(x) = \mathbb{E}_\xi[\Psi(x,\xi)]$, where $\xi$ is the random input of $\Psi(x,\xi)$ and $\psi(x)$ satisfies the following property:

$$\psi(x) - \psi(y) - \langle \psi'(y), x - y \rangle \leqslant \tfrac{L}{2}\left\|x - y\right\|^2 + M\left\|x - y\right\|.$$

Note that, when $M = 0$, function $\psi$ is Lipschitz smooth whereas when $L = 0$, it is nonsmooth. Due to the possible stochastic nature of $\Psi$, negative lower curvature in (6) and the combined smoothness and nonsmoothness property above, we have that $\psi$ can be either smooth or nonsmooth, deterministic or stochastic and convex ($\mu = 0$) or nonconvex ($\mu > 0$). We also assume bounded second moment stochastic oracle for $\psi'$ when $\psi$ is a stochastic function: For any $x$, we have an oracle whose output, $\Psi'(x,\xi)$, satisfies $\mathbb{E}_\xi[\Psi'(x,\xi)] = \psi'(x)$ and $\mathbb{E}[\|\Psi'(x,\xi) - \psi'(x)\|^2] \leqslant \sigma^2$.

For such a function, we consider an accelerated stochastic approximation algorithm (AC-SA) proposed in [15] for solving the subproblem (8) which can be reformulated as $\min_x \psi_k(x) + \mathbf{I}_{\{g_k(x) \leqslant \eta_k\}}(x)$, where $\mathbf{I}$ is the indicator set function. AC-SA algorithm can be applied when $\gamma \geqslant \mu$. In particular, $\psi_k(x) := \psi(x) + \tfrac{\gamma}{2}\left\|x - x^{k-1}\right\|^2$ is $(\gamma - \mu)$-strongly convex and $(L + \gamma)$-Lipschitz smooth. Moreover, AC-SA requires computation of a single prox operation of the following form in each iteration:

$$\operatorname*{argmin}_{x} w^T x + \left\|x - \bar{x}\right\|^2 + \mathbf{I}_{\{g_k(x) \leqslant \eta_k\}}(x), \tag{11}$$

for any $w, \bar{x} \in \mathbb{R}^d$. We show an efficient method for solving this problem at the end of in this section. For now, we look at convergence properties of the AC-SA:

**Proposition 3.6.** *[15] Let $x^k$ be the output of AC-SA algorithm after running $T_k$ iterations for the subproblem (8). Then $g_k(x^k) \leqslant \eta_k$ and $\mathbb{E}[\psi_k(x^k) - \psi_k(\bar{x}^k)] \leqslant \tfrac{2(L+\gamma)}{T_k^2}\left\|x^{k-1} - \bar{x}^k\right\|^2 + \tfrac{8(M^2+\sigma^2)}{(\gamma-\mu)T_k}$*

Note that convergence result in Proposition 3.6 closely follows the requirement in (9). In particular, we should ensure that $T_k$ is big enough such that $\tfrac{\rho}{2} \leqslant \tfrac{2(L+\gamma)}{T_k^2}$ and $\zeta_k = \tfrac{8(M^2+\sigma^2)}{(\gamma-\mu)T_k}$ sum to a constant. Consequently, we have the following corollary:

**Corollary 3.7.** *Let $\psi$ be nonconvex such that it satisfies (6) with $\mu > 0$. Set $\gamma = 3\mu$ and run AC-SA for $T_k = \max\{2\left(\tfrac{L}{\mu} + 3\right)^{1/2}, K(M + \sigma)\}$ iterations where $K$ is total iterations of Algorithm 1. Then, we obtain that $x^{\hat{k}}$ is an $(\varepsilon_1, \varepsilon_2)$-KKT point of (5), where $\hat{k}$ is chosen according to Theorem 3.5 and*

$$\varepsilon_1 = \left(\tfrac{3\Delta^0}{2K} + \tfrac{8(M+\sigma)}{\mu K}\right)\max\left\{\tfrac{8(9\mu^2+B^2L_h^2)}{\mu}, \tfrac{2BL_h}{\mu}\right\} + \tfrac{2B(\eta-\eta_0)}{K}, \quad \varepsilon_2 = \tfrac{3\Delta^0}{\mu K} + \tfrac{32(M+\sigma)}{\mu^2 K}$$

Note that Corollary 3.7 gives a unified complexity for obtaining KKT point of (5) in various settings of nonconvex objective ($\mu > 0$). First, in order to get an $\varepsilon$-KKT point, $K$ must be of $O(1/\varepsilon)$. If the problem is deterministic and smooth then $M = \sigma = 0$. In this case, $T_k = 2(\tfrac{L}{\mu} + 3)^{1/2}$ is a constant. Hence, the total iteration count is $\sum_{k=1}^K T_k = O(K)$, implying that total iteration complexity for obtaining an $\varepsilon$-KKT point is of $O(1/\varepsilon)$. For nonsmooth or stochastic cases, $M$ or $\sigma$ is positive. Hence, $T_k = O(K(M + \sigma))$ implying the total iteration complexity $\sum_{k=1}^K T_k = O(K^2)$, which is of $O(1/\varepsilon^2)$. Similar result for the convex case is shown in the appendix.

**Efficient projection**    We conclude this section by formally stating the theorem which provides an efficient oracle for solving the projection problem (11). Since $g_k(x) = \lambda\|x\|_1 + \langle v, x \rangle$, the linear form along with $\ell_1$ ball breaks the symmetry around origin which is used in existing results on (weighted) $\ell_1$-ball projection [10, 18]. Our method involves a careful analysis of Lagrangian duality equations to convert the problem into finding the root of a piecewise linear function. Then a line search method can be employed to find the solution in $O(d \log d)$ time. The formal statement is as follows:

**Theorem 3.8.** *There exists an algorithm that runs in $O(d \log d)$-time and solves the following problem exactly:*

$$\min_{x \in \mathbb{R}^d} \tfrac{1}{2}\left\|x - v\right\|^2 \text{ subject to } \|x\|_1 + \langle u, x \rangle \leqslant \tau. \tag{12}$$

In conclusion, note that (11) and (12) are equivalent where $v$ in (12) can be replaced by $\bar{x} + \tfrac{1}{2}w$ of (11) to get the equivalence of the objective functions of the two problems.

# 4 Experiments

The goal of this section is to illustrate the empirical performance of LCPP. For simplicity, we will consider the following learning problem:

$$\min_{x} \psi(x) = \tfrac{1}{n}\sum_{i=1}^{n} L_i(x), \quad \text{s.t.} \quad g(x) \leqslant \eta,$$

where $L_i(x)$ denotes the loss function. Specifically, we consider logistic loss $L_i(x) = \log(1 + \exp(-b_i a_i^T x))$ for classification and squared loss $L_i(x) = (b_i - a_i^T x)^2$ for regression. Here $(a_i, b_i)$ is the training sample, and $g(x)$ is the MCP penalty (see Table 2). Details of the testing datasets are summarized in Table 3. As we have stated, LCPP can be equipped with projected first order methods for fast iteration. We compare the efficiency of (spectral) gradient descent [16], Nesterov accelerated gradient and stochastic gradient [35] for solving LCPP subproblem. We find that spectral gradient outperforms the other methods in the logistic regression model and hence use it in LCPP for the remaining experiment for the sake of simplicity. Due to the space limit, we leave the discussion of this part in appendix. The rest of the section will compare the optimization efficiency of LCPP with the state-of-the-art nonlinear programming solver, and compare the proposed sparse constrained models solved by LCPP with standard convex and nonconvex sparse regularized models.

Table 3: Dataset description. R for regression and C for classification. `mnist` is formulated as a binary problem to classify digit 5 from the other digits. `real-sim` is randomly partitioned into 70% training data and 30% testing data.

| Datasets | Training size | Testing size | Dimensionality | Nonzeros | Types |
|---|---|---|---|---|---|
| `real-sim` | 50347 | 21962 | 20958 | 0.25% | C |
| `rcv1.binary` | 20242 | 677399 | 47236 | 0.16% | C |
| `mnist` | 60000 | 10000 | 784 | 19.12% | C |
| `gisette` | 6000 | 1000 | 5000 | 99.10% | C |
| `E2006-tfidf` | 16087 | 3308 | 150360 | 0.83% | R |
| `YearPredictionMSD` | 463,715 | 51,630 | 90 | 100% | R |

Our first experiment is to compare LCPP with existing optimization library for their optimization efficiency. To the best of our knowledge, DCCP ([27]) is the only open-source package available for the proposed nonconvex constrained problem. While the work [27] has made its code available online, we found that their code had unresolved errors in parsing MCP functions. Therefore, we replicate their setup in our own implementation. DCCP converts the initial problem into a sequence of relatively easier convex problems amenable to CVX ([9]), a convex optimization interface that runs on top of popular optimization libraries. We choose DCCP with MOSEK as the backend as it consistently outperforms DCCP with the default open-source solver SCS.

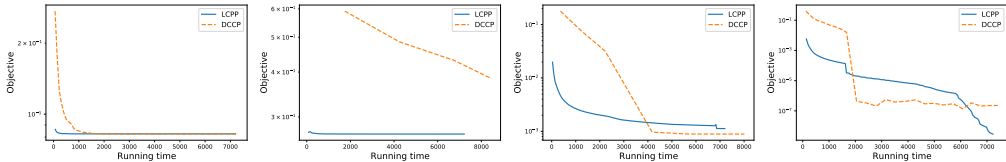

Figure 2: Objective value vs. running time (in seconds). Left to right: `mnist` ($\eta = 0.1d$), `real-sim` ($\eta = 0.001d$), `rcv1.binary` ($\eta = 0.05d$) and `gisette` ($\eta = 0.05d$). $d$ stands for the feature dimension.

Comparison is conducted on the classification problem. To fix the parameters, we choose $\gamma = 10^{-5}$ for `gisette` dataset and $\gamma = 10^{-4}$ for the other datasets. For each LCPP subproblem we run gradient descent at most 10 iterations and break when the criterion $\|x^k - x^{k-1}\|/\|x^k\| \leqslant \varepsilon$ is met. We set the number of outer loops as 1000 to run LCPP sufficiently long. We set $\lambda = 2, \theta = 0.25$ in the MCP function. Figure 2 plots the convergence performance of LCPP and DCCP, confirming that LCPP is more advantageous over DCCP. Specifically, LCPP outperforms DCCP, sometimes reaching

near-optimality even before DCCP finishes the first iteration. This observation can be explained by the fact that LCPP leverages the strengthen of first order methods, for which we can derive efficient projection subroutine. In contrast, DCCP is not scalable to large dataset due to the inefficiency in dealing with large scale linear system arising from the interior point subproblems.

Our next experiment is to compare the performance of nonconvex sparse constrained models, which is then optimized by LCPP, against regularized learning models in the following form:

$$\min_{x} \ \psi(x) = \frac{1}{n}\sum_{i=1}^{n} L_i(x) + \alpha g(x).$$

As described above, $g(x)$ is the sparsity-inducing penalty function and $L_i(x)$ is a loss function on the data. We consider both convex and nonconvex penalties, namely Lasso-type penalty $g(x) = \|x\|_1$ and MCP penalty (see Table 2). We solve the Lasso penalty problem by linear models provided by Sklearn [23] and solve the MCP regularized problem by the popular solver GIST [16]. For simplicity, both GIST and LCPP set $\lambda = 2$ and $\theta = 5$ in MCP function, and set the maximum iteration number as 2000 for all the algorithms. Then we use a grid of values $\alpha$ for GIST and LASSO, and $\eta$ for LCPP accordingly, to obtain the testing error under various sparsity levels. In Figure 3 we report the 0-1 error for classification and mean squared error for regression. We can clearly see the advantage of our proposed models over Lasso-type estimators. We observe that nonconvex models LCPP and GIST both perform more robustly than Lasso across a wide range of sparsity levels. Lasso models tend to overfit with increasing number of selected features while LCPP appears to be less affected by the feature selection.

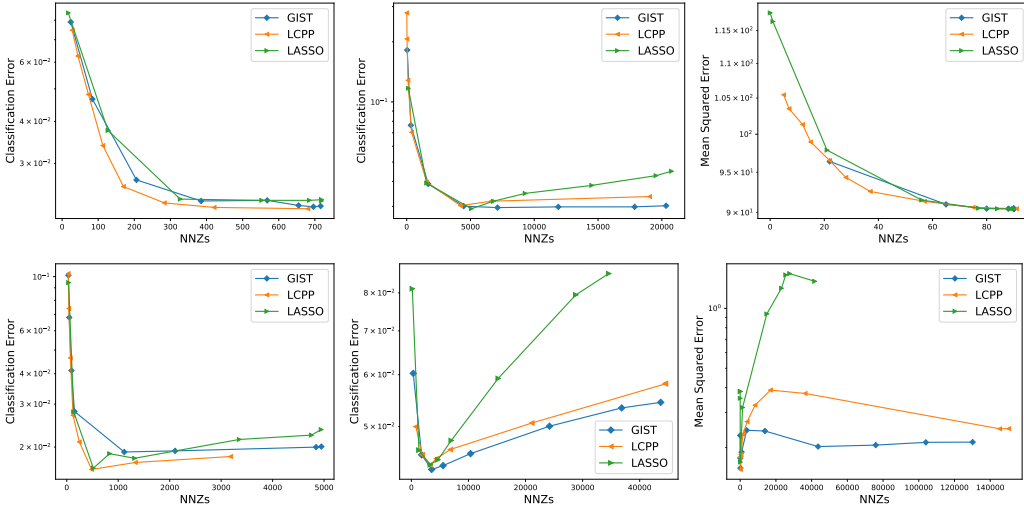

Figure 3: Testing error vs number of nonzeros. First two columns show classification performance in clockwise order: `mnist`, `real-sim`, `rcv1.binary` and `gisette`. The third column shows regression test on `YearPredictionMSD` (top) and `E2006` (bottom).

## 5 Conclusion

We present a novel proximal point algorithm (LCPP) for nonconvex optimization with a nonconvex sparsity-inducing constraint. We prove the asymptotic convergence of the proposed algorithm to KKT solutions under mild conditions. For practical use, we develop an efficient procedure for projection onto the subproblem constraint set, thereby adapting projected first order methods to LCPP for large-scale optimization and establish an $\mathcal{O}(1/\varepsilon)(\mathcal{O}(1/\varepsilon^2))$ complexity for deterministic (stochastic) optimization. Finally, we perform numerical experiments to demonstrate the efficiency of our proposed algorithm for large scale sparse learning.

## Broader Impact

This paper presents a new model for sparse optimization and performs an algorithmic study for the proposed model. A rigorous statistical study of this model is still missing. We believe this was due to the tacit assumption that constrained optimization was more challenging compared to regularized optimization. This work takes the first step in showing that efficient algorithms can be developed for the constrained model as well. Contributions made in this paper has the potential to inspire new research from statistical, algorithmic as well as experimental point of view in the wider sparse optimization area.

## Acknowledgments

Most of this work was done while Boob was at Georgia Tech. Boob and Lan gratefully acknowledge the National Science Foundation (NSF) for its support through grant CCF 1909298. Q. Deng acknowledges funding from National Natural Science Foundation of China (Grant 11831002).

## Footnotes

*Work done when author was at Georgia Tech

[2]Note that $\|\cdot\|_0$ is not a norm in mathematical sense. Indeed, $\|x\|_0 = \|tx\|_0$ for any nonzero $t$.

[3]Origin is always strictly feasible for sparsity inducing constraints and can be chosen as a starting point.

[4]Various subdifferentials exist in the literature for nonconvex optimization problem. Here, we use subdifferential Definition 3.1 in Boob et al. [5] for nonconvex nonsmooth function $g$.

[5]For specific examples of $g$, we show that origin is always the most feasible (and strictly feasible) solution of each subproblem and hence, does not require the predefined level-routine of LCPP to assert strict feasibility of subproblem. However, in order to keep generality of discussion, we perform the analysis under the level-setting.

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
