[Supplementary Material]

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

# A  Auxiliary results

## A.1  Existence of KKT points

**Proposition A.1.** *Under Assumption 3.1, let $x^0 = \hat{x}$. Then, for any $k \geqslant 1$, we have $x^{k-1}$ is strictly feasible for the $k$-th subproblem. Moreover, there exists $\bar{x}^k, \bar{y}^k \geqslant 0$ such that $g_k\left(\bar{x}^k\right) \leqslant \eta_k$ and:*

$$\partial \psi(\bar{x}^k) + \gamma\left(\bar{x}^k - x^{k-1}\right) + \bar{y}^k\left(\partial g_k(\bar{x}^k)\right) \ni 0 \tag{13}$$
$$\bar{y}^k\left(g_k\left(\bar{x}^k\right) - \eta_k\right) = 0$$

*Proof.* Since $x^0$ satisfies $g(x^0) \leqslant \eta_0 < \eta_1$ so we have that first subproblem is well defined. We prove the result by induction. First of all, suppose $x^{k-1}$ is strictly feasible for $k$-th subproblem: $g_k(x^{k-1}) < \eta_k$. Then we note that this problem is also valid and a feasible $x^k$ exists. Hence, algorithm is well-defined. Now, note that

$$g_{k+1}(x^k) = g(x^k) \leqslant g_k(x^k) \leqslant \eta_k < \eta_{k+1}.$$

where first inequality follows due to majorization, second inequality follows due to feasibility of $x^k$ for $k$-the subproblem and third strict inequality follows due to strictly increasing nature of sequence $\{\eta_k\}$.

Since $k$-th subproblem has $x^{k-1}$ as strictly feasible point satisfying Slater condition so we obtain existence of $\bar{x}^k$ and $\bar{y}^k \geqslant 0$ satisfying the KKT condition (13). ☐

## A.2  Proof of Theorem 3.3

In order to prove this theorem, we first state the following intermediate result.

**Proposition A.2.** *Let $\pi_k$ denotes the randomness of $x^1, x^2, ..., x^{k-1}$. Assume that there exists a $\rho \in [0, \gamma - \mu]$ and a summable nonnegative sequence $\zeta_k$ ($\zeta_k \geqslant 0$, $\sum_{k=1}^{\infty} \zeta_k < \infty$) such that*

$$\mathbb{E}\left[\psi_k(x^k) - \psi_k(\bar{x}^k)|\pi_k\right] \leqslant \frac{\rho}{2}\left\|\bar{x}^k - x^{k-1}\right\|^2 + \zeta_k \tag{14}$$

*Then, under Assumption 3.1, we have*
*1. The sequence $\mathbb{E}[\psi(x^k)]$ is bounded;*
*2. $\lim_{k \to \infty} \psi(x^k)$ exists a.s.;*
*3. $\lim_{k \to \infty} \left\|x^{k-1} - \bar{x}^k\right\| = 0$ a.s.;*
*4. If the whole algorithm is deterministic then $\psi(x^k)$ is bounded. Moreover, if $\zeta_k = 0$, then the sequence $\psi(x^k)$ is monotonically decreasing and convergent.*

*Proof.* Due to the strong convexity of $\psi_k(x)$, we have

$$\psi_k(\bar{x}^k) \leqslant \psi_k(x) - \frac{\gamma - \mu}{2}\left\|\bar{x}^k - x\right\|^2, \tag{15}$$

for all $x$ satisfying $g_k(x) \leqslant \eta_k$. Taking $x = x^{k-1}$ and using feasibility of $x^{k-1}$ ($g_k(x^{k-1}) \leqslant \eta_k$) we have

$$\psi(x^{k-1}) \geqslant \psi(\bar{x}^k) + \frac{\gamma}{2}\left\|\bar{x}^k - x^{k-1}\right\|^2 + \frac{\gamma - \mu}{2}\left\|x^{k-1} - \bar{x}^k\right\|^2$$

Together with (9) we have

$$\zeta_k + \psi(x^{k-1}) \geqslant \mathbb{E}\left[\psi(x^k) + \frac{\gamma}{2}\left\|x^k - x^{k-1}\right\|^2 |\pi_k\right] \tag{16}$$
$$+ \frac{\gamma - \mu - \rho}{2}\left\|x^{k-1} - \bar{x}^k\right\|^2.$$

Since $\{\zeta_k\}$ is summable, taking the expectation of $\pi_k$ and summing up all over all $k$, we have $\mathbb{E}[\psi(x^k)] \leqslant \psi(x^0) + \sum_{s=1}^{k}\zeta_k < \infty$. Moreover, Applying Supermartingale Theorem E.1 to (16), we have $\lim_{k \to \infty} \psi(x^k)$ exists and $\sum_{k=1}^{\infty}\left\|x^{k-1} - \bar{x}^k\right\|^2 < \infty$ a.s. Hence we conclude $\lim_{k \to \infty}\left\|x^{k-1} - \bar{x}^k\right\| = 0$ a.s. Part 4) can be readily deduced from (16). ☐

Now we are ready to prove Theorem 3.3.

For simplicity, we assume the whole sequence generated by Algorithm 1 converges to $\widetilde{x}$. Due to Proposition A.1, there exists a KKT point $(\bar{x}^k, \bar{y}^k)$. The optimality condition yields

$$\psi(x) + \frac{\gamma}{2}\left\|x - x^{k-1}\right\|^2 + \bar{y}^k g_k(x) \geqslant \psi(\bar{x}^k) + \frac{\gamma}{2}\left\|\bar{x}^k - x^{k-1}\right\|^2 + \bar{y}^k g_k(\bar{x}^k), \quad \forall x \qquad (17)$$

Since $\bar{y}^k$ is bounded, there exists a convergent subsequence $\{i_k\}$ that $\lim_{k\to\infty} \bar{y}^{i_k} = \widetilde{y}$ for some $\widetilde{y} \geqslant 0$. Let us take $k \to \infty$ in (17). In view of Proposition A.2, Part 3, we have $\lim_{k\to\infty} \bar{x}^{i_k} = \lim_{k\to\infty} x^{i_k-1} = \widetilde{x}$ almost surely. Then $\lim_{k\to\infty} h(x^{i_k-1}) = h(\widetilde{x})$ and $\lim_{k\to\infty} \nabla h(x^{i_k-1}) = \nabla h(\widetilde{x})$ a.s. due to the continuity of $h(x)$ and $\nabla h(x)$, respectively. Then we have

$$\psi(x) + \frac{\gamma}{2}\left\|x - \widetilde{x}\right\|^2 + \widetilde{y}\left[\lambda\left\|x\right\|_1 - h(\widetilde{x}) - \langle\nabla h(\widetilde{x}), x - \widetilde{x}\rangle\right] \geqslant \psi(\widetilde{x}) + \widetilde{y}g(\widetilde{x}), \quad a.s.$$

implying that $\widetilde{x}$ minimizes the loss function $\psi(x) + \frac{\gamma}{2}\left\|x - \widetilde{x}\right\|^2 + \widetilde{y}\left[\lambda\left\|x\right\|_1 - h(\widetilde{x}) - \langle\nabla h(\widetilde{x}), x - \widetilde{x}\rangle\right]$. Due to the first order optimality condition, we conclude $0 \in \partial\psi(\widetilde{x}) + \widetilde{y}\partial g(\widetilde{x})$, a.s.

Moreover, using the complementary slackness, we have $0 = \bar{y}^{i_k}\left(g_{i_k}\left(\bar{x}^{i_k}\right) - \eta_{i_k}\right)$. Taking the limit of $k \to \infty$ and noticing that $\lim_{k\to\infty}\eta_{i_k} = \eta$, we have $0 = \widetilde{y}\left(g\left(\widetilde{x}\right) - \eta\right)$ a.s . As a result, we conclude that $(\widetilde{x}, \widetilde{y})$ is a KKT point of problem (5), a.s.

## A.3 Proof of Theorem 3.4

From KKT condition of (13), $\bar{x}^k$ is the optimal solution of the problem $\min_{x\in\mathbb{R}^d}\psi_k(x) + \bar{y}^k\left(g_k(x) - \eta_k\right)$. Therefore, for any $x \in \mathbb{R}^d$, we have

$$\psi_k(x) + \bar{y}^k g_k(x) \geqslant \psi_k(\bar{x}^k) + \bar{y}^k g_k(\bar{x}^k) \qquad (18)$$

We prove that $\{\bar{y}^k\}$ is bounded a.s. by contradiction. If $\{\bar{y}^k\}$ has unbounded subsequence with positive probability, then conditioned under that event, there exists a subsequence $\{i_k\}$ such that $\bar{y}^{i_k} \to \infty$. Let us divide both sides of (18) by $\bar{y}^k$ and expand $g_k$ by its definition. After placing $k = i_k$, we have for all $x$

$$\begin{aligned}&\frac{1}{\bar{y}^{i_k}}\psi_{i_k}(x) + \lambda\left\|x\right\|_1 - \nabla h(x^{i_k-1})^T x\\ &\geqslant \frac{1}{\bar{y}^{i_k}}\psi_{i_k}(\bar{x}^{i_k}) + \lambda\left\|\bar{x}^{i_k}\right\|_1 - \nabla h(x^{i_k-1})^T\bar{x}^{i_k}.\end{aligned} \qquad (19)$$

Let $\widetilde{x}$ be any limiting point a.s. of the sequence $\left\{x^{i_k-1}\right\}$. By the statement of the theorem, we know that it exists and satisfies MFCQ assumption. Passing to some subsequence if necessary, we have $\lim_{k\to\infty}x^{i_k-1} = \widetilde{x}$ a.s. Using Proposition A.2 Part 3, we have $\lim_{k\to\infty}\bar{x}^{i_k} = \widetilde{x}$ a.s. Moreover, using Proposition A.2 Part 2, we have $\lim_{k\to\infty}\psi(\bar{x}^{i_k})$ exists a.s. This implies $\lim_{k\to\infty}\frac{1}{\bar{y}^{i_k}}\psi_{i_k}(\bar{x}^{i_k}) = 0$ a.s.

Taking $k \to \infty$, since $\psi_{i_k}(x)$ is bounded a.s. (due to existence of $\widetilde{x}$ a.s.), we have $\lim_{k\to\infty}\frac{1}{\bar{y}^{i_k}}\psi_{i_k}(x) = 0$. From Lipschitz continuity of $l_1$ norm and $\nabla h(x)$, we have $\lim_{k\to\infty}\lambda\left\|\bar{x}^{i_k}\right\|_1 = \lambda\left\|\widetilde{x}\right\|_1$ a.s., and $\lim_{k\to\infty}\nabla h(x^{i_k-1}) = \nabla h(\widetilde{x})$ a.s., respectively. It then follows from (19) that for all $x$, we have $\lambda\left\|x\right\|_1 - \langle\nabla h(\widetilde{x}), x\rangle \geqslant \lambda\left\|\widetilde{x}\right\|_1 - \langle\nabla h(\widetilde{x}), \widetilde{x}\rangle$. In other words, we have

$$\mathbf{0} \in \partial\lambda\left\|\widetilde{x}\right\|_1 - \nabla h(\widetilde{x}) = \partial g(\widetilde{x}), a.s. \qquad (20)$$

Moreover, due to complementary slackness and $\bar{y}^{i_k} > 0$, the equality $g_{i_k}(\bar{x}^{i_k}) = \eta_{i_k}$ holds. Hence, in the limit, we have the constraint $g(\widetilde{x}) = \eta$ active a.s. Under MFCQ, there exists $z$ such that $\max_{v\in\partial g(\widetilde{x})}z^T v < 0$. However, from (20) we have $0 = z^T\mathbf{0}$ since $\mathbf{0} \in \partial g(\widetilde{x})$, leading to a contradiction to the event that $\{\bar{y}^k\}$ contained unbounded sequence with positive probability. Hence, $\bar{y}$ is bounded a.s.

## B Explicit and specialized bounds on the dual

Here, we discuss some of the results for explicit bounds on the dual. In particular, we focus on the SCAD and MCP case. Similar results can be extended for Exp and $\ell_p, p < 0$ case since these function follows two key properties (as we will see later in the proofs):

1. $|\nabla h(x)| \leqslant \lambda$ for all $x$ for each of these functions.
2. They remain bounded below a constant. See Figure 1.

We exploit these two structural properties of these sparse constraints to obtain specialized and explicit bounds on the optimal dual of problem 5. The following lemma is in order.

**Lemma B.1.** *Let $h : \mathbb{R} \to \mathbb{R}$ be the the convex function which satisfies $|\nabla h(x)| \leqslant \lambda$ for all $x \in \mathbb{R}$. Then the minimum value of $\bar{g}(x; \bar{x}) : \mathbb{R} \to \mathbb{R}$ defined as $\bar{g}(x; \bar{x}) := \lambda|x| - h(\bar{x}) - \langle \nabla h(\bar{x}), x - \bar{x} \rangle$ is achieved at $0$ for all $\bar{x} \in \mathbb{R}$.*

*Proof.* Note that $\bar{g}$ is a convex function for any $\bar{x} \in \mathbb{R}$. So by first order optimality condition, if $\hat{x}$ is the minimizer of $\bar{g}$ then $0 \in \partial \bar{g}(\hat{x}; \bar{x})$. This implies

$$\lambda \partial |\hat{x}| - \nabla h(\bar{x}) \ni 0.$$

Note that $\hat{x} = 0$ satisfies this condition since in that case $\lambda \partial |\hat{x}| = [-\lambda, \lambda]$. And due to assumption on $h$, we have $\nabla h(\bar{x}) \in [-\lambda, \lambda]$. Hence $\hat{x} = 0$ is always the minimizer. $\square$

Now note that $h_{\lambda,\theta}$ functions defined for our examples, such as SCAD or MCP. satisfy the assumption of bounded gradients in Lemma B.1. Now we use this simple result to show that $\mathbf{0}$ is the most feasible solution for each of the subproblem (8) generated in Algorithm 1 and hence we can give an explicit bound for the optimal dual value for each subproblem.

**Lemma B.2.** *Suppose all assumptions in Lemma B.1 are satisfied. Then we have for any $k \geqslant 1$,*

$$\bar{y}^k \leqslant \frac{\psi_k(\mathbf{0}) - \psi_k(\bar{x}^k)}{\eta_k - g(x^{k-1}) + \sum_{i=1}^d (\lambda - |\nabla h(x_i^{k-1})|)|x_i^{k-1}|}. \tag{21}$$

*Proof.* Note that $g_k(x) = \sum_{i=1}^d \bar{g}(x_i; x_i^{k-1})$ where $\bar{g}$ is defined in Lemma B.1. Since assumptions of Lemma B.1 hold, so we have that each individual $\bar{g}$ is minimized at $x_i = 0$. Hence $g_k(\mathbf{0})$ is the minimum value of $g_k$. In view of Proposition A.1, we have that $x_{k-1}$ is strictly feasible solution with respect to constraint $g_k(x) \leqslant \eta_k$ implying $g_k(x^{k-1}) - \eta_k < 0$. Hence, we have

$$\begin{aligned}
&\eta_k - g_k(\mathbf{0}) \\
&= \eta_k - \left[ \lambda \|\mathbf{0}\|_1 - \sum_{i=1}^d \{h(x_i^{k-1}) + \nabla h(x_i^{k-1})(0 - x_i^{k-1})\} \right] \\
&= \eta_k + \sum_{i=1}^d h(x_i^{k-1}) - \sum_{i=1}^d \nabla h(x_i^{k-1}) x_i^{k-1} \\
&= \eta_k - g(x^{k-1}) + [g(x^{k-1}) + h(x^{k-1})] - \sum_{i=1}^d \nabla h(x_i^{k-1}) x_i^{k-1} \\
&\geqslant \eta_k - g(x^{k-1}) + \lambda \|x^{k-1}\|_1 - \sum_{i=1}^d |\nabla h(x_i^{k-1})||x_i^{k-1}| \\
&= \eta_k - g(x^{k-1}) + \sum_{i=1}^d (\lambda - |\nabla h(x_i^{k-1})|)|x_i^{k-1}| \\
&> 0.
\end{aligned}$$

Here, last strict inequality follows due to the fact that $\lambda \geqslant |\nabla h(x_i^{k-1})|$ and $\eta_k > g(x^{k-1})$. Then, we have, optimal dual $\bar{y}^k$ satisfies for all $x$:

$$\begin{aligned}
\psi_k(\bar{x}^k) &\leqslant \psi_k(x) + \bar{y}^k(g_k(x) - \eta_k) \\
\Rightarrow \psi_k(\bar{x}^k) &\leqslant \psi_k(\mathbf{0}) + \bar{y}^k(g_k(\mathbf{0}) - \eta_k) \\
\Rightarrow \bar{y}^k &\leqslant \frac{\psi_k(\mathbf{0}) - \psi_k(\bar{x}^k)}{\eta_k - g_k(\mathbf{0})} \\
&\leqslant \frac{\psi_k(\mathbf{0}) - \psi_k(\bar{x}^k)}{\eta_k - g(x^{k-1}) + \sum_{i=1}^d (\lambda - |\nabla h(x_i^{k-1})|)|x_i^{k-1}|},
\end{aligned}$$

where third inequality follows due to the fact that $\eta_k - g_k(\mathbf{0}) > 0$ Hence, we conclude the proof. $\square$

Note that the bound in (21) depends on $x^{k-1}$ which can not be controlled, especially in the stochastic cases. In order to show a bound on $\bar{y}^k$ irrespective of $x^{k-1}$, we must lower bound the denominator in (21) for all possible values of $x^{k-1}$. To accomplish this goal, we show the following two theorems in which we lower bound the term $\sum_{i=1}^d (\lambda - |\nabla h(x_i^{k-1})|)|x_i^{k-1}|$. Each of these theorem is a specialized result for SCAD and MCP function, respectively.

Figure 4: Plot of $z(\gamma)$ for SCAD function where $\lambda = 1$, $\theta = 5$. $z : [0,3] \to \mathbb{R}_{\geqslant 0}$ where $z(0) = z(3) = 0$ otherwise $z$ is strictly positive.

**Theorem B.3.** *Let $g$ be the SCAD function and $x \in \mathbb{R}^d$ such that $g(x) = \alpha$. Also, let $\gamma = \alpha - \beta \frac{\lambda^2(\theta+1)}{2}$ where $\beta$ is the largest nonnegative integer such that $\gamma \geqslant 0$. Then, $\sum_{i=1}^d (\lambda - |\nabla h(x_i)|)|x_i| \geqslant z(\gamma)$ where $z : [0, \frac{\lambda^2(\theta+1)}{2}] \to \mathbb{R}_{\geqslant 0}$ is the function defined as*

$$z(\gamma) := \begin{cases} \gamma & \text{if } 0 \leqslant \gamma \leqslant \lambda^2 \\ \frac{\gamma}{\lambda}\sqrt{\frac{2}{\theta-1}}\sqrt{\frac{\lambda^2(\theta+1)}{2} - \gamma} & \text{if } \lambda^2 < \gamma \leqslant \frac{\lambda^2(\theta+1)}{2} \end{cases} .$$

**Theorem B.4.** *Let $g$ be the MCP function and $x \in \mathbb{R}^d$ be such that $g(x) = \alpha$. Also let $\gamma = \alpha - \beta \frac{\lambda^2\theta}{2}$ where $\beta$ is the largest nonnegative integer such that $\gamma \geqslant 0$. Then $\sum_{i=1}^d (\lambda - |\nabla h(x_i)|)|x_i| \geqslant z(\gamma)$ where $z : [0, \frac{\lambda^2\theta}{2}] \to \mathbb{R}_{\geqslant 0}$ is the function defined as $z(\gamma) := \gamma\sqrt{1 - \frac{2\gamma}{\theta\lambda^2}}$.*

Note that Theorem B.3 states that lower bound $z(\gamma) = 0$ when $\gamma = 0$ or $\frac{\lambda^2(\theta+1)}{2}$. In essence, when $\alpha$ is exact integral multiple of $\frac{\lambda^2(\theta+1)}{2}$ then lower bound turn out to be zero. However, for all other values of $\alpha$, the corresponding $z(\gamma)$ is strictly positive. This can be seen from the graph of $z(\gamma)$ below. Similar claims can be made with respect to MCP in Theorem B.4.

Now we are ready to show a bound on $\bar{y}^k$ irrespective of $x^{k-1}$. We give a specific routine to choose the values of $\eta_k$ such that we can obtain a provable bound on the denominator in (21) hence obtaining an upper bound on the $\bar{y}^k$ for all $k$ irrespective of $x^{k-1}$.

**Proposition B.5.** *Let $g$ be the SCAD function and $\eta = \beta \frac{\lambda^2(\theta+1)}{2} + \tilde{\eta}$ where $\beta$ be the largest nonnegative integer such that $\tilde{\eta} \geqslant 0$. Then, for properly selected $\eta_0$, we have that $\eta_k - g(x^{k-1}) + \sum_{i=1}^d (\lambda - |\nabla h(x_i^{k-1})|)|x_i^{k-1}| \geqslant \min\{\lambda^2, \frac{z(\tilde{\eta})}{2}\}$.*

We note that very similar proposition for MCP can be proved based on Theorem B.4. We skip that discussion in order to avoid repetition.

**Connection to MFCQ**    In this section, we show the connection of MFCQ assumption in Theorem 3.4 with the bound in Theorem B.3.

Note that for the boundary points of the set $g(x) \leqslant \eta_1$ where $\eta_1 = \frac{\lambda^2(\theta+1)}{2}$ then the lower bound $z(\eta_1) = 0$. In fact, carefully following the proof of Theorem B.3, we can identify that the lower bound is tight for $x$'s such that one of the coordinate $x_i$ satisfy $|x_i| \geqslant \lambda\theta$ and all other coordinates are 0. In this case, we see that such points do not satisfy MFCQ. At such points, we don't have any strictly feasible directions required by MFCQ assumption. This can be easily visualized in the Figure 5 part (a) below. Note that $\lambda\theta = 5$ and for any $|x| \geqslant 5$, the feasible region is merely the axis and hence there is no strict feasible direction. This implies MFCQ indeed fails at these points.

For $g(x) = \eta_2 < \eta_1$ the lower bound $z(\eta_2)$ is nonzero and same holds for $g(x) = \eta_3 > \eta_1$. Indeed, we see that for such cases, the points not satisfying MFCQ in case of $\eta_1$ vanish. This can be observed in Figure 5 part (b) and part (c). For the case of $\eta_2$ in part (b), these points become infeasible and for the case of $\eta_3$ in part (c), they are no longer boundary points.

Looking back at MFCQ from the result of Theorem B.3, we can see that how close $\eta$ is to $\frac{\lambda^2(\theta+1)}{2}$ shows how 'close' the problem is for violating MFCQ. Moreover, the lower bound $z(\cdot)$ on the denominator of (21) shows how quickly the dual will explode as the problem setting gets closer to violating MFCQ.

(a) $\eta_1 = 3$        (b) $\eta_2 = 2.8$        (c) $\eta_3 = 3.2$

Figure 5: All figures are plotted for $\lambda = 1$ and $\theta = 5$. Then $\eta_1 = \frac{\lambda^2(\theta+1)}{2} = 3$. In fig (a), we see that for $|x| \geqslant 5$, the MFCQ assumption is violated since only $x$-axis is feasible. Similar observation holds for $y$-axis as well. However, in fig(b) and fig(c) such claims are no longer valid.

We complete this discussion by showing the proof of Theorem B.3 and Theorem B.4. We also note that similar theorems can be proved for $\ell_p, p < 0$ and Exp function in Table 2.

## B.1 Proof of Theorem B.3

First, we show a lower bound for one-dimensional function and then extend it to higher dimensions. Suppose $u \in \mathbb{R}$ be such that $g(u) = \alpha$. Note that since $g$ is SCAD function so $\alpha$ must lie in the set $[0, \frac{\lambda^2(\theta+1)}{2}]$. Key to our analysis is the lower bound on $(\lambda - |\nabla h(u)|)|u|$ as a function of $\alpha$. Note that since

$$g(u) = \alpha \Rightarrow \lambda|u| \geqslant \alpha \Rightarrow |u| \geqslant \tfrac{\alpha}{\lambda}. \tag{22}$$

Also note that for all $|u| \leqslant \lambda$, we have $g(u) = \lambda|u|$ and $\nabla h(u) = 0$ which implies $\nabla h(u) = 0$ for all $g(u) = \alpha \leqslant \lambda^2$. Hence, using this relation along with (22), we obtain

$$(\lambda - |\nabla h(u)|)|u| = \lambda|u| \geqslant \alpha \qquad \text{if } 0 \leqslant \alpha \leqslant \lambda^2. \tag{23}$$

We note that $|\nabla h(u)| = \lambda$ for all $u \geqslant \lambda\theta$ and $g(u) = \alpha = \frac{\lambda^2(\theta+1)}{2}$ for all $u \geqslant \lambda\theta$. Hence,

$$(\lambda - |\nabla h(u)|)|u| = 0 \qquad \text{if } \alpha = \tfrac{\lambda^2(\theta+1)}{2}. \tag{24}$$

Now we design a lower bound when $\alpha \in (\lambda^2, \frac{\lambda^2(\theta+1)}{2})$. For such values of $\alpha$, we have

$$\begin{aligned}
g(u) &= \lambda|u| - \tfrac{(|u|-\lambda)^2}{2(\theta-1)} = \alpha \\
&\Rightarrow u^2 - 2\lambda\theta|u| + \lambda^2 + 2\alpha(\theta-1) = 0 \\
&\Rightarrow |u| = \lambda\theta - \sqrt{2(\theta-1)\big[\tfrac{\lambda^2(\theta+1)}{2} - \alpha\big]} \\
&\Rightarrow |\nabla h(u)| = \tfrac{|u|-\lambda}{\theta-1} = \lambda - \sqrt{\tfrac{2}{\theta-1}}\sqrt{\tfrac{\lambda^2(\theta+1)}{2} - \alpha} \\
&\Rightarrow \lambda - |\nabla h(u)| = \sqrt{\tfrac{2}{\theta-1}}\sqrt{\tfrac{\lambda^2(\theta+1)}{2} - \alpha}.
\end{aligned}$$

Then, above relation along with (22), we have $(\lambda - |\nabla h(u)|)|u| \geqslant \sqrt{\frac{2}{\theta-1}}\frac{\alpha}{\lambda}\sqrt{\frac{\lambda^2(\theta+1)}{2} - \alpha}$ for all $\alpha \in (\lambda^2, \frac{\lambda^2(\theta+1)}{2})$. Using this relation along with (23), (24) and noting the definition of function $z(\cdot)$, we obtain a lower bound $(\lambda - |\nabla h(u)|)|u| \geqslant z(\alpha)$ where $\alpha = g(u)$.

Now note that for general high-dimensional $x \in \mathbb{R}^d$, we have $g(x) = \sum_{i=1}^d g(x_i) = \alpha$. Then $\alpha \in [0, \frac{d\lambda^2(\theta+1)}{2}]$. Since each individual $g(x_i) \geqslant 0$, we can think of $\alpha$ as a budget such that sum of

$g(x_i)$ must equal $\alpha$. In order to minimize the lower bound on $(\lambda - |\nabla h(x_i)|)|x_i|$, we should exhaust the largest budget from $\sum_{i=1}^{d} g(x_i) = \alpha$ while maintaining the lowest possible value of the lower bound on $(\lambda - |\nabla h(x_i)|)|x_i|$. This clearly holds by setting $|x_i|$ such that $g(x_i) = \frac{\lambda^2(\theta+1)}{2}$. This can be clearly observed in the figure below.

Figure 6: Plot of function $z(\alpha)$ on $y$-axis and $\alpha$ on $x$-axis for $\lambda = 1$, $\theta = 5$. The largest possible value $g(u)$ is $\frac{\lambda^2(\theta+1)}{2} = 3$ is achieved for $u \geqslant \lambda\theta = 5$ and lower bound $z(3) = 0$. Hence, setting $u \geqslant \lambda\theta$ maximizes the $g(u)$ and minimizes $z(\alpha) = z(g(u))$.

Hence, if $\alpha \in \left[\beta\frac{\lambda^2(\theta+1)}{2}, (\beta+1)\frac{\lambda^2(\theta+1)}{2}\right)$ for some nonnegative integer $\beta$, then we should set $\beta$ coordinates of $x$ satisfying $|x_i| \geqslant \lambda\theta$ in order to exhaust the maximum possible budget, $\frac{\lambda^2(\theta+1)}{2}$, from $\alpha$ and still keep the value of the lower bound on $(\lambda - |\nabla h(u)|)|u|$ as 0. Hence, noting the definition of $\gamma$, the problem reduces to $\sum_i g(x_i) = \gamma$ where summation is taken over remaining coordinates of $x$ and $\gamma \in \left[0, \frac{\lambda^2(\theta+1)}{2}\right)$.

Lets recall from the analysis in 1-D case that if $g(x_i) = \alpha_i$ then $(\lambda - |\nabla h(x_i)|)|x_i| \geqslant z(\alpha_i)$ so we obtain the lower bound $\sum_i z(\alpha_i)$ while $\alpha_i$'s satisfy the relation $\sum_i \alpha_i = \gamma$. Moreover, $z : [0, \frac{\lambda^2(\theta+1)}{2}] \to \mathbb{R}_{\geqslant 0}$ is a concave function with $z(0) = 0$. Then we show that $z$ is a subadditive function. Using Jensen's inequality, for all $t \in [0, 1]$, we have $z(tx + (1-t)y) \geqslant tz(x) + (1-t)z(y)$. Using $y = 0$ and the fact that $z(0) = 0$, we have $z(tx) \geqslant tz(x)$ for any $t \in [0, 1]$. Now using this relation along with $t = \frac{x}{x+y} \in [0, 1]$ (for $x, y \geqslant 0$) we have

$$z(x) = z(t(x+y)) \geqslant tz(x+y).$$
$$z(y) = z((1-t)(x+y)) \geqslant (1-t)z(x+y).$$

Adding the two relations, we obtain $z(x) + z(y) \geqslant z(x+y)$. Hence, $z$ is a subadditive function. Since $\sum_i \alpha_i = \gamma$ then the we have $\sum_i z(\alpha_i) \geqslant z(\sum_i \alpha_i) = z(\gamma)$. This bound is indeed achieved when we set one of $\alpha_i = \gamma$ and rest to 0. Hence, we conclude the proof.

## B.2  Proof of Theorem B.4

As before, we proceed by assuming 1-D case, i.e., $u \in \mathbb{R}$ and $g(u) = \alpha$ and then extend it to general $d$-dimensional setting. Then, $\alpha \in [0, \frac{\lambda^2\theta}{2}]$. Then, we write function $(\lambda - |\nabla h(u)|)|u|$ in term of $\alpha$. Note that

$$g(u) = \lambda|u| - \frac{u^2}{2\theta} = \alpha$$
$$\Rightarrow |u| = \theta\lambda\left(1 - \sqrt{1 - \frac{2\alpha}{\theta\lambda^2}}\right)$$
$$\Rightarrow |\nabla h(u)| = \frac{|u|}{\theta} = \lambda\left(1 - \sqrt{1 - \frac{2\alpha}{\theta\lambda^2}}\right)$$
$$\Rightarrow \lambda - |\nabla h(u)| = \lambda\sqrt{1 - \frac{2\alpha}{\theta\lambda^2}}$$

Moreover, we also have (22). Then, noting the definition of $z(\cdot)$, we obtain that $(\lambda - |\nabla h(u)|)|u| \geqslant z(\alpha)$.

For high dimensional $x \in \mathbb{R}^d$, we use similar arguments as in the proof of theorem B.3. In particular, we set $\beta$ coordinates $x$ satisfying $|x_i| \geqslant \lambda\theta$ which exhausts the maximum possible budget $\frac{\lambda^2\theta}{2}$ from $\alpha$ and still keeps the value of the lower bound on $(\lambda - |\nabla h(x_i)|)|x_i|$ as 0. Finally, we reduce the problem to $\sum_i g(x_i) = \sum_i \alpha_i = \gamma$ and lower bound is $\sum_i z(\alpha_i)$. As in the previous case, $z$ is concave function on nonnegative domain with $z(0) = 0$ hence it must be subadditive. So we obtain that $\sum_i z(\alpha_i) \geqslant z(\sum_i \alpha_i) = z(\gamma)$. Hence, we conclude the proof.

## B.3 Proof of Proposition B.5

We note that $\eta = \beta\frac{\lambda^2(\theta+1)}{2} + \widetilde{\eta}$, where $\beta$ is the largest nonnegative integer such that $\widetilde{\eta} \geqslant 0$. Clearly $\widetilde{\eta} \in \left[0, \frac{\lambda^2(\theta+1)}{2}\right)$. Now, we divide our analysis in two cases:

**Case 1:** Suppose $\widetilde{\eta} \leqslant \lambda^2$. Then we define $\eta_0$ for Algorithm 1 as $\eta_0 = \beta\frac{\lambda^2(\theta+1)}{2} + \frac{\widetilde{\eta}}{2}$.

Now, if $g(x^{k-1}) \leqslant \beta\frac{\lambda^2(\theta+1)}{2}$ then we have that $\eta_{k-1} - g(x^{k-1}) \geqslant \eta_0 - g(x^{k-1}) \geqslant \frac{\widetilde{\eta}}{2}$. In this case, we obtain that denominator of (21) is at least $\frac{\widetilde{\eta}}{2}$.

In other case, suppose that $g(x^{k-1}) > \beta\frac{\lambda^2(\theta+1)}{2}$. We also note that $g(x^{k-1}) \leqslant g_{k-1}(x^{k-1}) \leqslant \eta_{k-1} \leqslant \eta$. Hence, we obtain $g(x^{k-1}) \leqslant \eta = \beta\frac{\lambda^2(\theta+1)}{2} + \widetilde{\eta}$. This implies $\widetilde{g}(x^{k-1}) := g(x^{k-1}) - \beta\frac{\lambda^2(\theta+1)}{2} \in [0, \lambda^2]$. Then, using Theorem B.3, we obtain that $\sum_{i=1}^d (\lambda - |\nabla h(x_i^{k-1})|)|x_i^{k-1}| \geqslant z(\widetilde{g}(x^{k-1})) = \widetilde{g}(x^{k-1})$. Using this relation, we obtain that $\eta_{k-1} - g(x^{k-1}) + \sum_{i=1}^d (\lambda - |\nabla h(x_i^{k-1})|)|x_i^{k-1}| \geqslant \eta_{k-1} - g(x^{k-1}) + \widetilde{g}(x^{k-1}) = \eta_{k-1} - \beta\frac{\lambda^2(\theta+1)}{2} = \widetilde{\eta}_{k-1} \geqslant \frac{\widetilde{\eta}}{2}$.

So, when $\widetilde{\eta} \leqslant \lambda^2$, we obtain that the denominator in (21) is at least $\eta_k - \eta_{k-1} + \frac{z(\widetilde{\eta})}{2} = \delta_k + \frac{z(\widetilde{\eta})}{2} \geqslant \frac{z(\widetilde{\eta})}{2}$.

**Case 2:** Now, we look at the second case where $\widetilde{\eta} > \lambda^2$. In this case, we define $\eta_0 = \beta\frac{\lambda^2(\theta+1)}{2} + \min\{\lambda^2, z(\widetilde{\eta})\}$. Then, we again note that $g(x^{k-1}) \leqslant \beta\frac{\lambda^2(\theta+1)}{2}$ implies $\eta_{k-1} - g(x^{k-1}) \geqslant \widetilde{\eta}_{k-1} \geqslant \widetilde{\eta}_0$.

In other case, we assume that $g(x^{k-1}) \in [\beta\frac{\lambda^2(\theta+1)}{2}, \beta\frac{\lambda^2(\theta+1)}{2} + \lambda^2]$, then again using Theorem B.3, we obtain $\sum_{i=1}^d (\lambda - |\nabla h(x_i^{k-1})|)|x_i^{k-1}| \geqslant z(\widetilde{g}(x^{k-1})) = \widetilde{g}(x^{k-1})$. This implies $\eta_{k-1} - g(x^{k-1}) + \sum_{i=1}^d (\lambda - |\nabla h(x_i^{k-1})|)|x_i^{k-1}| \geqslant \eta_{k-1} - \beta\frac{\lambda^2(\theta+1)}{2} = \widetilde{\eta}_{k-1} \geqslant \widetilde{\eta}_0$.

Finally, $g(x^{k-1}) > \beta\frac{\lambda^2(\theta+1)}{2} + \lambda^2$ then $\widetilde{g}(x^{k-1}) \in (\lambda^2, \widetilde{\eta})$ then due to concavity of $z$, we obtain that $z(\widetilde{g}(x^{k-1})) \geqslant \min\{\lambda^2, z(\widetilde{\eta})\} = \widetilde{\eta}_0$.

Hence, combining the bounds in both cases, we obtain that denominator in (21) is always bounded below by $\min\{\lambda^2, \frac{z(\eta)}{2}\}$.

## C Proof of Theorem 3.5

As in the previous case, we show an important recursive property of iterates. We first state the theorem again:

**Theorem C.1.** *Suppose Assumption 3.1, 3.2 hold such that $\delta_k = \frac{\eta - \eta_0}{k(k+1)}$ for all $k \geqslant 1$. Let $\pi_k$ denote the randomness of $x^1, \ldots, x^{k-1}$. Suppose for $k$-th subproblem (8), the solution $x^k$ satisfies*

$$\mathbb{E}[\psi_k(x^k) - \psi_k(\bar{x}^k)|\pi_k] \leqslant \frac{\rho}{2}\left\|x^{k-1} - \bar{x}^k\right\|^2 + \zeta_k,$$

$$g_k(x^k) \leqslant \eta_k$$

*where $\rho$ lies in the interval $[0, \gamma - \mu]$ and $\{\zeta_k\}$ is a sequence of nonnegative numbers. If $\hat{k}$ is chosen uniformly randomly from $\lfloor\frac{K+1}{2}\rfloor$ to $K$ then corresponding to $x^{\hat{k}}$, there exists pair $(\bar{x}^{\hat{k}}, \bar{y}^{\hat{k}})$ satisfying*

$$\mathbb{E}_{\hat{k}}\left[\text{dist}\left(\partial_x\mathcal{L}(\bar{x}^{\hat{k}}, \bar{y}^{\hat{k}}), 0\right)^2\right] \leqslant \frac{8(\gamma^2 + B^2 L_h^2)}{K(\gamma - \mu - \rho)}\left(\frac{\gamma - \mu + \rho}{\gamma - \mu}\Delta^0 + 2Z_1\right),$$

$$\mathbb{E}_{\hat{k}}\left[\bar{y}^{\hat{k}}|g(\bar{x}^{\hat{k}}) - \eta|\right] \leqslant \frac{2BL_h}{K(\gamma - \mu - \rho)}\left(\frac{\gamma - \mu + \rho}{\gamma - \mu}\Delta^0 + 2Z_1\right) + \frac{2B(\eta - \eta_0)}{K},$$

$$\mathbb{E}_{\hat{k}}\left\|x^{\hat{k}} - \bar{x}^{\hat{k}}\right\|^2 \leqslant \frac{4\rho(\gamma - \mu + \rho)}{K(\gamma - \mu)^2(\gamma - \mu - \rho)}\Delta^0 + \frac{8Z_1}{K(\gamma - \mu - \rho)},$$

where, $\Delta^0 := \psi(x^0) - \psi(x^*)$ and $Z_1 := \sum_{k=1}^{K} \zeta_k$.

We first prove the following important relationship on the sum of squares of distances of the iterates.

**Proposition C.2.** *Let requirements of Theorem 3.5 hold. Then for any $s \geqslant 2$, we have*

$$\mathbb{E}[\sum_{k=s}^{K} \|x^{k-1} - \bar{x}^k\|^2 |\pi_{s-1}] \leqslant \frac{2(A_s + Z_s)}{\gamma - \mu - \rho}, \tag{25}$$

$$\mathbb{E}[\sum_{k=s}^{K} \|x^k - \bar{x}^k\|^2 |\pi_{s-1}] \leqslant \frac{2\rho A_s}{(\gamma - \mu)(\gamma - \mu - \rho)} + \frac{2Z_s}{\gamma - \mu - \rho} \tag{26}$$

*where $A_s = \frac{\gamma - \mu + \rho}{\gamma - \mu} \left[ \psi(x^{s-2}) - \psi(x^*) \right]$ and $Z_s = \sum_{k=s-1}^{K} \zeta_k$.*

*Proof.* Note that since for all $k \geqslant 1$ we have feasibility of $x^k$ for $k$-th subproblem (due to (10)), then in view of Proposition A.1, we have that $x^{k-1}$ is strictly feasible for the $k$-th subproblem. Consequently, using strong convexity of $\psi_k$ and optimality of $\bar{x}^k$, we have $\frac{\gamma - \mu}{2} \|x^{k-1} - \bar{x}^k\|^2 \leqslant \psi_k(x^{k-1}) - \psi_k(\bar{x}^k)$. Therefore, taking expectation conditioned on $\pi_{k-1}$ ob both sides of the above relation, we obtain

$$\frac{\gamma - \mu}{2} \mathbb{E}[\|x^{k-1} - \bar{x}^k\|^2 |\pi_{k-1}] \leqslant \mathbb{E}[\psi_k(x^{k-1}) - \psi_k(\bar{x}^k)|\pi_{k-1}]$$
$$\leqslant \mathbb{E}[\psi_{k-1}(x^{k-1}) - \psi_k(\bar{x}^k)|\pi_{k-1}]$$
$$\leqslant \psi_{k-1}(\bar{x}^{k-1}) - \mathbb{E}[\psi_k(\bar{x}^k)|\pi_{k-1}] + \frac{\rho}{2} \|x^{k-2} - \bar{x}^{k-1}\|^2 + \zeta_{k-1}$$

where second inequality follows from $\psi_k(x^{k-1}) = \psi(x^{k-1}) \leqslant \psi_{k-1}(x^{k-1})$ and third inequality follows from (9). Placing the definition of $\psi_k(\cdot)$ in above relation, we have

$$\frac{2\gamma - \mu}{2} \mathbb{E}[\|x^{k-1} - \bar{x}^k\|^2 |\pi_{k-1}] \leqslant \psi(\bar{x}^{k-1}) - \mathbb{E}[\psi(\bar{x}^k)|\pi_{k-1}] + \frac{\gamma + \rho}{2} \|x^{k-2} - \bar{x}^{k-1}\|^2 + \zeta_{k-1}.$$

Summing up over $k = s, s+1, \ldots, K$ and taking expectation conditioned on $\pi_{s-1}$, we have

$$\frac{2\gamma - \mu}{2} \sum_{k=s}^{K} \mathbb{E}[\|x^{k-1} - \bar{x}^k\|^2 |\psi_{s-1}] \leqslant \psi(\bar{x}^{s-1}) - \mathbb{E}\psi(\bar{x}^K)$$
$$+ \frac{\gamma + \rho}{2} \sum_{k=s}^{K} \mathbb{E}[\|x^{k-2} - \bar{x}^{k-1}\|^2 |\pi_{s-1}] + \sum_{k=s}^{K} \zeta_{k-1}.$$

It then follows that

$$\frac{\gamma - \mu - \rho}{2} \mathbb{E}\left[\sum_{k=s}^{K} \|x^{k-1} - \bar{x}^k\|^2 |\pi_{s-1}\right] \leqslant \psi(\bar{x}^{s-1}) - \mathbb{E}\psi(\bar{x}^K) + \frac{\gamma + \rho}{2} \|x^{s-2} - \bar{x}^{s-1}\|^2 + \sum_{k=s}^{K} \zeta_{k-1}$$
$$\leqslant \psi_{s-1}(\bar{x}^{s-1}) - \mathbb{E}\psi(\bar{x}^K)$$
$$+ \frac{\rho}{\gamma - \mu} \left[\psi_{s-1}(x^{s-2}) - \psi_{s-1}(\bar{x}^{s-1})\right] + \sum_{k=s}^{K} \zeta_{k-1}$$
$$\leqslant \psi(x^{s-2}) - \mathbb{E}\psi(\bar{x}^K)$$
$$+ \frac{\rho}{\gamma - \mu} \left[\psi(x^{s-2}) - \psi_{s-1}(\bar{x}^{s-1})\right] + \sum_{k=s}^{K} \zeta_{k-1}$$
$$\leqslant \frac{\gamma - \mu + \rho}{\gamma - \mu} \left[\psi(x^{s-2}) - \psi(x^*)\right] + \sum_{k=s}^{K} \zeta_{k-1},$$

where the third and the last inequality follow from the property

$$\psi(x^{k-1}) = \psi_k(x^{k-1}) \geqslant \psi_k(\bar{x}^k) \geqslant \psi(\bar{x}^k) \geqslant \psi(x^*).$$

Note that solution $x^k$ is feasible for the $k$-th subproblem and hence, in view of Proposition A.1, we have that $g(\bar{x}^k) \leqslant g_k(\bar{x}^k) \leqslant \eta_k < \eta$ and hence $\bar{x}^k$ is feasible solution for the main problem implying $\psi(\bar{x}^k) \geqslant \psi(x^*)$ in the above relation. Then (25) immediately follows.

Now we prove that (26) holds. Note that

$$\mathbb{E}\left[\|x^k - \bar{x}^k\|^2 |\pi_k\right] \leqslant \frac{2}{\gamma - \mu} \mathbb{E}\left[\psi_k(x^k) - \psi_k(\bar{x}^k)|\pi_k\right] \leqslant \frac{2}{\gamma - \mu} \left[\frac{\rho}{2} \|x^{k-1} - \bar{x}^k\|^2 + \zeta_k\right],$$

where the first inequality follows due to the strong convexity $\psi_k$ as well as the optimality of $\bar{x}^k$ and the second inequality follows due to (9). Now summing the above relation from $k = s$ to $K$ and taking expectation conditioned on $\psi_{s-1}$, we obtain

$$\mathbb{E}\left[\sum_{k=s}^{K} \|x^k - \bar{x}^k\|^2 |\pi_{s-1}\right] \leqslant \frac{\rho}{\gamma - \mu} \mathbb{E}\left[\sum_{k=s}^{K} \|x^{k-1} - \bar{x}^k\|^2 |\pi_{s-1}\right] + \frac{2}{\gamma - \mu} \sum_{k=s}^{K} \zeta_k$$
$$\leqslant \frac{2\rho A_s}{(\gamma - \mu)(\gamma - \mu - \rho)} + \frac{2Z_s}{\gamma - \mu - \rho},$$

where the last inequality follows from (25) and the definition of $Z_s$. Hence, we conclude the proof. $\square$

Now we present the unified convergence of proximal point as stated in Theorem 3.5.

*Proof of Theorem 3.5.* Due to the KKT condition for the subproblem (8), we have

$$0 \in \partial\psi(\bar{x}^k) + \gamma\left(\bar{x}^k - x^{k-1}\right) + \bar{y}^k\left(\partial\left\|\bar{x}^k\right\|_1 - \nabla h(x^{k-1})\right)$$
$$0 = \bar{y}^k\left(\lambda\left\|\bar{x}^k\right\|_1 - h(x^{k-1}) - \left\langle\nabla h(x^{k-1}), \bar{x}^k - x^{k-1}\right\rangle - \eta_k\right) \tag{27}$$

Using triangle inequality along with first relation in the above equation, we have $\text{dist}\left(\partial_x\mathcal{L}(\bar{x}^k, \bar{y}^k), 0\right) \leqslant \gamma\left\|\bar{x}^k - x^{k-1}\right\| + \bar{y}^k\left\|\nabla h(x^{k-1}) - \nabla h(\bar{x}^k)\right\|$. Therefore, noting the bound on $\bar{y}^k$ from Assumption 3.2, we have

$$\text{dist}\left(\partial_x\mathcal{L}(\bar{x}^k, \bar{y}^k), 0\right)^2 \leqslant 2\gamma^2\left\|\bar{x}^k - x^{k-1}\right\|^2 + 2B^2\left\|\nabla h(x^{k-1}) - h(\bar{x}^k)\right\|^2$$
$$\leqslant 2\left(\gamma^2 + B^2 L_h^2\right)\left\|\bar{x}^k - x^{k-1}\right\|^2,$$

where the second inequality uses Lipschitz smoothness of $h(x)$. Summing the above relation from $k = s, \ldots, K$ and the taking expectation conditioned on $\pi_{s-1}$ on both sides, we obtain

$$\mathbb{E}\left[\sum_{k=s}^K \text{dist}\left(\partial_x\mathcal{L}(\bar{x}^k, \bar{y}^k), 0\right)^2 \Big| \pi_{s-1}\right] \leqslant 2(\gamma^2 + B^2 L_h^2)\mathbb{E}\left[\sum_{k=s}^K\left\|x^{k-1} - \bar{x}^k\right\|^2 \Big| \pi_{s-1}\right]$$
$$\leqslant \frac{4\left(\gamma^2 + B^2 L_h^2\right)}{\gamma - \mu - \rho}(A_s + Z_s), \tag{28}$$

For the complementary slackness part of the KKT condition, first notice that $\eta_k = \eta_0 + \sum_{t=1}^k \delta_t = \eta_0 + \sum_{t=1}^k \frac{\eta - \eta_0}{t(t+1)} = \frac{k}{k+1}\eta + \frac{1}{k+1}\eta_0$. Therefore,

$$\sum_{k=s}^K (\eta - \eta_k) = \sum_{k=s}^K \frac{\eta - \eta_0}{k+1} \leqslant \frac{K+1-s}{s+1}(\eta - \eta_0).$$

To prove the error of complementary slackness condition, observe that

$$\bar{y}^k\left|\lambda\left\|\bar{x}^k\right\|_1 - h(\bar{x}^k) - \eta\right| \leqslant \bar{y}^k\left|\lambda\left\|\bar{x}^k\right\|_1 - h(x^{k-1}) - \left\langle\nabla h(x^{k-1}), \bar{x}^k - x^{k-1}\right\rangle - \eta_k\right|$$
$$+ \bar{y}^k\left|h(x^{k-1}) + \left\langle\nabla h(x^{k-1}), \bar{x}^k - x^{k-1}\right\rangle - h(\bar{x}^k)\right| + \bar{y}^k(\eta - \eta_k)$$
$$\leqslant \frac{BL_h}{2}\left\|\bar{x}^k - x^{k-1}\right\|^2 + B(\eta - \eta_k),$$

where second inequality follows due to second relation in (27) and bound on $\bar{y}^k$ from Assumption 3.2. Summing the above relation from $k = s, \ldots, K$ and taking expectation conditioned on $\pi_{s-1}$ on both sides, we obtain

$$\mathbb{E}\left[\sum_{k=s}^K \bar{y}^k\left|g(\bar{x}^k) - \eta\right| \Big| \pi_{s-1}\right] \leqslant \sum_{k=s}^K \mathbb{E}\left[\frac{BL_h}{2}\left\|\bar{x}^k - x^{k-1}\right\|^2 + B(\eta - \eta_k) \Big| \pi_{s-1}\right]$$
$$\leqslant \frac{BL_h}{2}\mathbb{E}\left[\sum_{k=s}^K\left\|\bar{x}^k - x^{k-1}\right\|^2 \Big| \psi_{s-1}\right] + B\sum_{k=s}^K(\eta - \eta_k)$$
$$\leqslant \frac{BL_h}{\gamma - \mu - \rho}(A_s + Z_s) + \frac{(K+1-s)B(\eta - \eta_0)}{s+1}. \tag{29}$$

Now note that $A_s = \frac{\gamma - \mu + \rho}{\gamma - \mu}[\psi(x^{s-2}) - \psi(x^*)]$ is a random variable due to randomness of $x^{s-2}$. Now we bound expectation of $\psi(x^{s-2})$. In view of (9), we have

$$\mathbb{E}[\psi_k(x^k)|\pi_k] \leqslant \psi_k(\bar{x}^k) + \frac{\rho}{2}\left\|x^{k-1} - \bar{x}^k\right\|^2 + \zeta_k$$
$$\leqslant \psi_k(x^{k-1}) - \frac{\gamma - \mu - \rho}{2}\left\|x^{k-1} - \bar{x}^k\right\| + \zeta_k$$

Since, $\gamma - \mu - \rho \geqslant 0$ and noting that $\psi_k(x^{k-1}) = \psi(x^{k-1})$, $\psi_k(x^k) \geqslant \psi(x^k)$, we have

$$\mathbb{E}[\psi(x^k)|\pi_k] \leqslant \psi(x^{k-1}) + \zeta_k.$$

Taking expectation on both sides of the above relation and then summing from $k = 1$ to $s - 2$, we get

$$\mathbb{E}[\psi(x^{s-2})] \leqslant \psi(x^0) + \sum_{k=1}^{s-2}\zeta_k.$$

Using the above relation, we obtain

$$\mathbb{E}[A_s] \leqslant \frac{\gamma - \mu + \rho}{\gamma - \mu}\Delta^0 + 2\sum_{k=1}^{s-2}\zeta_k, \tag{30}$$

where $\Delta^0 = \psi(x^0) - \psi(x^*)$. Note that here we used the fact $\frac{\gamma-\mu+\rho}{\gamma-\mu} \leqslant 2$. Now taking expectation on both sides of (28) and using bound on $\mathbb{E}[A_s]$ in (30), we obtain

$$\mathbb{E}\left[\sum_{k=s}^K \text{dist}\left(\partial_x \mathcal{L}(\bar{x}^k, \bar{y}^k), 0\right)^2 \big| \pi_{s-1}\right] \leqslant \frac{4(\gamma^2+B^2L_h^2)}{\gamma-\mu-\rho}\left(\frac{\gamma-\mu+\rho}{\gamma-\mu}\Delta^0 + 2\sum_{k=1}^{s-2}\zeta_k + \sum_{k=s-1}^K \zeta_k\right)$$
$$\leqslant \frac{4(\gamma^2+B^2L_h^2)}{\gamma-\mu-\rho}\left(\frac{\gamma-\mu+\rho}{\gamma-\mu}\Delta^0 + 2Z_1\right).$$

Similarly, taking expectation on both sides of (29) and using (30), we obtain

$$\mathbb{E}\left[\sum_{k=s}^K \bar{y}^k \left|g(\bar{x}^k)-\eta\right| \big| \pi_{s-1}\right] \leqslant \frac{BL_h}{\gamma-\mu-\rho}\left(\frac{\gamma-\mu+\rho}{\gamma-\mu}\Delta^0 + 2Z_1\right) + \frac{K+1-s}{s+1}B(\eta-\eta_0).$$

Taking expectation on both sides of (26) and using (30), we obtain

$$\mathbb{E}[\sum_{k=s}^K \|x^k - \bar{x}^k\|^2] \leqslant \frac{2\rho}{(\gamma-\mu)(\gamma-\mu-\rho)}\left(\frac{\gamma-\mu+\rho}{\gamma-\mu}\Delta^0 + 2\sum_{k=1}^{s-2}\zeta_k\right) + \frac{2Z_s}{\gamma-\mu-\rho}$$
$$\leqslant \frac{2\rho(\gamma-\mu+\rho)}{(\gamma-\mu)^2(\gamma-\mu-\rho)}\Delta^0 + \frac{4Z_1}{\gamma-\mu-\rho}.$$

Finally, setting $s = \lfloor \frac{K+1}{2} \rfloor$, we have $\frac{K}{2} \leqslant s \leqslant \frac{K+1}{2}$. Therefore, we have

$$\mathbb{E}_{\hat{k}}\left[\text{dist}\left(\partial_x \mathcal{L}(\bar{x}^{\hat{k}}, \bar{y}^{\hat{k}}), 0\right)^2\right] \leqslant \frac{8(\gamma^2+B^2L_h^2)}{K(\gamma-\mu-\rho)}\left(\frac{\gamma-\mu+\rho}{\gamma-\mu}\Delta^0 + 2Z_1\right),$$

$$\mathbb{E}_{\hat{k}}\left[\bar{y}^{\hat{k}}\left|g(\bar{x}^{\hat{k}})-\eta\right|\right] \leqslant \frac{2BL_h}{K(\gamma-\mu-\rho)}\left(\frac{\gamma-\mu+\rho}{\gamma-\mu}\Delta^0 + 2Z_1\right) + \frac{2B(\eta-\eta_0)}{K},$$

and

$$\mathbb{E}_{\hat{k}}\left\|x^{\hat{k}} - \bar{x}^{\hat{k}}\right\|^2 \leqslant \frac{4\rho(\gamma-\mu+\rho)}{K(\gamma-\mu)^2(\gamma-\mu-\rho)}\Delta^0 + \frac{8Z_1}{K(\gamma-\mu-\rho)}.$$

Hence, we conclude the proof. $\qquad\square$

## C.1 Proof of Corollary 3.7

Since $T_k \geqslant 2\sqrt{\frac{L}{\mu}+3}$, we have that $\frac{2(L+\gamma)}{T_k^2} = \frac{2(L+3\mu)}{T_k^2} \leqslant \frac{\mu}{2} = \frac{\rho}{2}$. Moreover, we see that $\rho = \mu \leqslant \gamma - \mu = 2\mu$. Finally, since $T_k \geqslant K(M+\sigma)$ so we have $\zeta_k \leqslant \frac{4}{\mu K}$ implying that $Z_1 = \sum_{k=1}^K \zeta_k \leqslant \frac{4}{\mu}$. Then, applying Theorem 3.5, we obtain that $x^{\hat{k}}$ is an $(\varepsilon_1, \varepsilon_2)$-KKT solution of the problem (5).

## C.2 Convergence for the (stochastic) convex case

We have the following Corollary of Theorem 3.5 for the case in which objective $\psi$ is convex, i.e. $\mu = 0$.

**Corollary C.3.** *Let $\psi$ be convex function such that it satisfies* (6) *with $\mu = 0$. Set $\gamma = \beta L$ where $\beta \in [0,1)$ be a small constant and run AC-SA for $T_k = \max\{2\sqrt{\frac{2(1+\beta)}{\beta}}, K(M+\sigma)\}$ iterations where $K$ is the total number of iterations of Algorithm 1. Then, $x^{\hat{k}}$ is an $(\varepsilon_1, \varepsilon_2)$-KKT point of the problem* (5) *where*

$$\varepsilon_1 = \left(\frac{3\Delta^0}{2K} + \frac{16(M+\sigma)}{\beta KL}\right)\max\left\{\frac{16(\beta^2L^2+B^2L_h^2)}{\beta L}, \frac{4BL_h}{\beta L}\right\} + \frac{2B(\eta-\eta_0)}{K},$$

$$\varepsilon_2 = \frac{3\Delta^0}{2\beta LK} + \frac{128(M+\sigma)}{\beta L^2 K}.$$

*Proof.* Since $T_k \geqslant 2\sqrt{\frac{2(1+\beta)}{\beta}}$, we have $\frac{2(L+\gamma)}{T_k^2} = \frac{2(1+\beta)L}{T_k^2} \leqslant \frac{\beta L}{4} = \frac{\rho}{2}$. Moreover, note that $\rho = \frac{\beta L}{2} \leqslant \gamma = \beta L$. Finally, since $T_k \geqslant K(M+\sigma)$ so we have $\zeta_k = \frac{8(M^2+\sigma^2)}{\gamma T_k} \leqslant \frac{8(M+\sigma)}{\beta LK}$. Hence, $Z_1 = \sum_{k=1}^K \zeta_k \leqslant \frac{8(M+\sigma)}{\beta L}$. Then, applying Theorem 3.5, we obtain that $x^{\hat{k}}$ is an $(\varepsilon_1, \varepsilon_2)$-KKT solution of problem (5). $\qquad\square$

**Finite-sum problem** A special case of objective takes the finite-sum form $f(x) = \frac{1}{n}\sum_{i=1}^{n}\widetilde{f}_i(x)$ thereby leading to the following subproblem

$$\min_x \widetilde{\psi}(x) = \frac{1}{n}\sum_{i=1}^{n}\widetilde{f}_i(x) + \widetilde{\omega}(x)$$

It is known that finite-sum problem can be efficiently solved by using variance reduction or randomized incremental gradient method [35, 20]. The complexity of LCPP on finite-sum problem can be further improved if we apply variance reduction technique for solving the subproblem. We comment on the complexity result in brief. In the finite-sum setting, the Nesterov's accelerated gradient-based LCPP requires $T_k = \widetilde{\mathcal{O}}(n\sqrt{\frac{L+2\mu}{\mu}})$ and $T_k = \widetilde{\mathcal{O}}(n\beta^{-1/2})$ number of stochastic gradient computations to solve each LCPP subproblem. Even though this number is a constant in terms of dependence on $K$, number of terms $(n)$ in the finite sum can be large. In comparison to these standard methods, the complexity of SVRG (stochastic variance reduced gradient) based LCPP method can be improved to $T_k = \widetilde{\mathcal{O}}(n + \frac{L+\mu}{\mu})$ for the case when $\psi$ is nonconvex satisfying (6) with $\mu > 0$, and to $T_k = \widetilde{\mathcal{O}}(n + \beta^{-1})$ for convex problem where $\mu = 0$.

## D Proof for the projection algorithm for problem (11)

We formulate the update as the following problem

$$\min_{x\in\mathbb{R}^d} \frac{1}{2}\|x - v\|^2 \text{ s.t. } \|x\|_1 + \langle u, x\rangle \leqslant \tau. \tag{31}$$

Since the objective is strongly convex, problem (31) has a unique global optimal solution. Moreover, the problem is strictly feasible because of the strict feasibility guarantee (A.1) in the context of problem (8). Therefore, KKT condition guarantees that there exists $y \geqslant 0$ such that

$$0 \in x - v + yu + y\partial\|x\|_1, \tag{32}$$
$$0 = y\left(\langle u, x\rangle + \|x\|_1 - \tau\right). \tag{33}$$

The algorithm proceeds as follows. First, we check whether $v$ is feasible, if it is the case, then $x = v$ is the optimal solution. Otherwise, the constraint in (31) is active. Next, we explore the optimality condition (32). Given the optimal Lagrangian multiplier $y \geqslant 0$, for the $i$-th coordinate of the optimal $x$, one of the following three situations will occur:

1. $x_i > 0$ and $x_i = v_i - (u_i + 1)y$.
2. $x_i < 0$ and $x_i = v_i - (u_i - 1)y$.
3. $x_i = 0$ and $(u_i - 1)y \leqslant v_i \leqslant (u_i + 1)y$.

For simplicity, let us denote $[a]_+ = \max\{a, 0\}$ and $[a, b]_+ = \max\{a, b, 0\}$. Based on the discussion above, we can express $x$ as a piecewise linear function of $y$.

$$x_i(y) = [v_i - (u_i + 1)y]_+ - [(u_i - 1)y - v_i]_+.$$

Let us denote $\ell(y) = \langle u, x(y)\rangle + \|x(y)\|_1$. We can deduce that

$$\begin{aligned}
\ell(y) &= \sum_{i=1}^{d} u_i x_i(y) + \sum_{i=1}^{d} \max\{x_i(y), -x_i(y)\} \\
&= \sum_{i=1}^{d} u_i [v_i - (u_i + 1)y]_+ - \sum_{i=1}^{d} u_i [(u_i - 1)y - v_i]_+ \\
&\quad + 2\sum_{i=1}^{d} [v_i - (u_i + 1)y, (u_i - 1)y - v_i]_+ \\
&\quad - \sum_{i=1}^{d} [v_i - (u_i + 1)y]_+ - \sum_{i=1}^{d} [(u_i - 1)y - v_i]_+ \\
&= \sum_{i=1}^{d} (u_i - 1) [v_i - (u_i + 1)y]_+ \\
&\quad - \sum_{i=1}^{d} (u_i + 1) [(u_i - 1)y - v_i]_+ \\
&\quad + 2\sum_{i=1}^{d} [v_i - (u_i + 1)y, (u_i - 1)y - v_i]_+
\end{aligned}$$

Above, the second equality uses the identity: $\max\{p - q, q - p\} = 2\max\{p, q\} - p - q$ for any $p, q \in \mathbb{R}$. It can be readily seen that $\ell(y)$ is a piecewise linear function with at most $3d$ breaking points. We can sort these points in $\mathcal{O}(d\log d)$ and then apply a line-search to find the root of $\ell(\cdot) = \tau$ in $\mathcal{O}(d)$ time.

# E  Supermartingale convergence theorem

In below, we state a version of supermartingale convergence theorem developed by [25].

**Theorem E.1.** *Let $(\Omega, F, P)$ be a probability space and $\mathcal{F}_0 \subseteq \mathcal{F}_1 \subseteq ... \subseteq \mathcal{F}_k \subseteq$ be some sub-$\sigma$-algebra of $F$. Let $b_k$, $c_k$ be nonnegative $\mathcal{F}_k$-measurable random variables such that*

$$\mathbb{E}\left[b_{k+1} \mid \mathcal{F}_k\right] \leqslant b_k + \xi_k - c_k,$$

*where $\{\xi_k\}_{0 \leqslant k < \infty}$ is a non-negative and summable: $\sum_{k=0}^{\infty} \xi_k < +\infty$. Then we have*

$$\lim_{k \to \infty} b_k \text{ exists, and } \sum_{k=1}^{\infty} c_k < +\infty, \quad a.s.$$

# F   Additional experiments

This section describes additional experiments for investigating the empirical performance of LCPP. We run all the algorithms on a cluster node with Intel Xeon Gold 2.6G CPU and 128G RAM.

**Solving the subproblems**

We compare the performance of different instances of LCPP for which the subproblems are solved by a variety of convex algorithms. Specifically, we consider LCPP-SVRG, LCPP-SGD, LCPP-NAG and LCPP-BB in which the subproblems are solved by proximal stochastic variance reduced gradient descent (SVRG [35]), proximal stochastic gradient descent (SGD), Nesterov's accelerated gradient (NAG[22]) and spectral gradient (Barzilai-Borwein stepsize) respectively. We adopt the spectral gradient descent with non-monotone line search from [16] due to its superior performance in the reported experiments.

Figure 7: Objective value vs. number of effective passes over the dataset. Green, orange, blue and red curves represent NAG, SGD, SVRG and BB. We set $\eta = \alpha d$. First row: `gisette` ($\alpha = 0.05, 0.10$, left to right); second row: rcv1.binary, ($\alpha = 0.10, 0.20$), third row: real-sim ($\alpha = 0.10, 0.20$).

Figure 7 shows the objective vs. number of effective passes over the datasets. Here, each effective pass evaluates one full gradient. We find that stochastic algorithms (LCPP-SGD, LCPP-SVRG) converge more rapidly than deterministic algorithms (LCPP-NAG, LCPP-BB) in the earlier stage, but they do not obtain higher accuracy in the long run. In all the tested datasets, we can observe that LCPP-BB outperforms the other three methods. Moreover, we remark that stochastic gradient algorithms need to compute projections more frequently than deterministic algorithms. While our linesearch routine

can efficiently perform projection, it is still more expensive than computing stochastic gradient, particularly, for the sparse data. Hence the overall running time of SGD algorithms is much worse than that of LCBB-BB. For the above reasons, we choose LCPP-BB as our default choice in the main experiment section.

**Classification performance**

We conduct an additional experiment to compare the empirical performance of all the tested algorithms in sparse logistic regression. We perform grid search based on five-fold cross-validation to find the best hyper-parameters. Then we retrain each model with the chosen hyper-parameter on the whole training dataset and report the classification performance on the testing data. Each experiment is repeated five times. Hyper-parameters: 1) GIST: $\alpha = 1$, $n\lambda \in \{10, 1, 0.1\}$ where $n$ is the size of training data, $\theta \in \{100, 10, 5, 1, 0.1, 0.01, 0.001\}$, 2) LCPP: $\lambda = 2$, $\theta \in \{100, 10, 5, 1, 0.1, 0.01, 0.001\}$, $\eta = 10^{-k}d$ where $k \in \{-3, -2.5, -2, -1.5, -1\}$, 3) Lasso: we set $C = C_0 10^s$ where $s = 1 + \frac{2}{3}k$, $k = 0, 1, 2, ..., 9$, and $C_0$ is chosen by the l1_min_c function in Sklearn. Table 4 summarizes the testing performance (mean and standard deviation) of each compared method. We can observe from this table that LCPP achieves the best performance on three out of the four datasets.

Table 4: Classification error (%) of different methods for sparse logistic regression

| Datasets | GIST | LCPP | LASSO |
|---|---|---|---|
| gisette | $2.32 \pm 0.04$ | $\mathbf{1.64 \pm 0.14}$ | $1.84 \pm 0.05$ |
| mnist | $2.57 \pm 0.01$ | $\mathbf{2.52 \pm 0.02}$ | $2.56 \pm 0.00$ |
| rcv1.binary | $6.39 \pm 0.03$ | $4.90 \pm 0.14$ | $\mathbf{4.52 \pm 0.01}$ |
| realsim | $3.50 \pm 0.04$ | $\mathbf{3.03 \pm 0.00}$ | $3.10 \pm 0.00$ |