[Reviews · NeurIPS 2020]

Review 1

Summary and Contributions: ============== UPDATE AFTER REBUTTAL ===================== Thank you for your response. The authors have agreed to clarify the motivation for the use of non-convex constraints and will be precise about the use of the word "suboptimal". They have agreed to state that they have considerably more knowledge on the specific kind of non-convexity they are dealing with when comparing with prior works. As a consequence I have increased my score, although I believe the experimental section remains rather unconvincing. ============================================== This paper proposes a method based on a sequence of convex approximations to solve optimization problems with a non-convex sparsity constraint. As stated by the authors, their approach is a special case of existing frameworks, notably part I of [25]. The authors use standard techniques such as adding a proximal term to the objective and convexifying the constraint by exploiting its DC structure to derive the convex subproblems. Their contribution lies in the derivation and the use of projection-based algorithms to solve the convex subproblems. This guarantees the strict feasibility of the sequence of solutions of the subproblems, and ultimately leads to better complexity bounds than existing black-box non-convex algorithms which convexify by adding proximal terms and solve subproblems with penalty methods.

Strengths: There is broad interest in problems with spasity constraints so the work is relevant to the NeurIPS community. The authors carefully adapt non-convex optimization techniques to the special case of having a sparsity constraint whose structure is well known (i.e part 2 of Assumption 2.1 and line 180). They exploit this added knowledge of the constraint to derive a projection algorithm as well as better convergence results which rely on the strict feasibility of solutions of subproblems. In this sense, the work is novel.

Weaknesses: The arguments for the use of non-convex approximations of the \ell_0 norm over the \ell_1 norm are unconvincing in their current form. For instance, can the authors explain the use of 'suboptimal' on line 30? We know that under minor technical conditions, Lasso is minimax optimal. Moreover, the experimental section doesn't show much evidence that the convex LASSO is indeed beaten by non-convex approaches. Furthermore, the authors could be more upfront about the assumptions they make on g earlier within the introduction.

Correctness: Detailed proofs of the Theorems are provided in the Appendix. They seem correct to me.

Clarity: The paper is well written and well organized. The sequence of theorems and assumptions is natural and pleasant. The authors could however properly define\bar{x}, \bar{y} and \bar{x}^k etc whenever they're used (as they don't seem to refer to a KKT point every time, see for instance (11)). For example, including line 437 in the main text instead of late in the appendix, would really help the reader.

Relation to Prior Work: There is a good effort made in discussing prior work. Relevant works are summarised well. However, the authors could be more fair to works which tackle black-box non-convex constraints. Their work has considerably more information on the non-convex constraint, a fact which is exploited in several of their main results. So the authors should state that their gains in complexity are coming from the fact that they have more information on g. There are other existing works that combine knowledge of hard sparsity and convex constraints, such as Kyrillidis et al (CLASH). Their scheme uses an alternating projection algorithm to the non-convex set first, followed by the convex projection, which does not hurt since it is non-expansive. Note that this example is derived only for quadratic functions with RIP but nonetheless it should serve as a good comparison basis point, which is missed.

Reproducibility: Yes

Additional Feedback: In Figure 3 in the experimental section, if my understanding is correct, it seems pretty clear that having no sparsity constraint is better for all datasets but one (rcv1.binary). Could the authors explain why they opted for logistic regression on these datasets ?(and not for some inverse problems for example) In section 4, I think it is good practice to cite the datasets you use. Line 634 : small typo on the conditioning in the second inequality. Line 40 : Problem (5) does fit within existing frameworks for non-convex optimisation, so could the authors clarify why they believe (5) wasn't studied solely because of the algorithmic difficulty ? Can you please expand on the benefits of (5) over (4) ? The following is a suggestion for improvement : An algorithm for solving non-convex sparsity constraint on the rank of matrices would be interesting, do the authors see any way of extending their work to the approximations of rank instead of \ell_0 norm of vectors ?


Review 2

Summary and Contributions: This paper proposes a feasible level proximal point method for minimizing a smooth objective function subject to a DC constraint. The proposed method works by solving a sequence of convex subproblems with gradually relaxed constraint levels. For the sparsity constraint problem, the subproblem can be solved exactly by a fast break points search algorithm within O(nlog(n)). The authors also provide the convergence and convergence rate of the algorithm, even when the objective is smooth/nonsmooth and deterministic/stochastic. Some numerical experiments on large scale sparse optimization problems have been conducted to demonstrate the effectiveness of the proposed algorithm.

Strengths: S1. The authors propose a new level proximal point method with gradually increasing the relaxed constraint levels. The convergence results seem reasonable and interesting. S2. The authors provide the convergence and convergence rate of the algorithm for both deterministic and stochastic settings. S3. The authors provide sufficient details of the projection algorithm for the subproblem and the numerical experiments.

Weaknesses: W1. The motivation from the perspective of sparse optimization is weak. There are some efficient and effective sparse optimization methods such as greedy pursuit method, DC penalized method, coordinate-wise optimization method, and block-k decomposition method that can solve the sparse optimization problem. The authors do not compare with these methods to achieve exact sparse signal recovery in their experiments. W2. The algorithm needs to gradually increase the parameter eta, which could slow down the convergence of the algorithm. Moreover, the algorithm involves some tricky parameters eta (Note that one can add a constant to both side of the inequality of (5) to change the value of eta) and eta0, which could significantly impact the performance of the algorithm. The use of the parameter eta is similar to the diminishing step size strategy for updating the primal/dual variables as in subgradient descent algorithm and non-convex ADMM algorithm. W3. The comparisons in the experiments may not be enough. The authors should compare the proposed algorithm with the operator splitting algorithms such as alternating direction method of multipliers.

Correctness: I think the proof are reasonable.

Clarity: Yes

Relation to Prior Work: Yes

Reproducibility: No

Additional Feedback: I have read the rebuttal and other reviewers' comments. I am satisfied with the authors' responses. Thus, I stick to my score.


Review 3

Summary and Contributions: After rebuttal: I am satisfied with the authors responses to my questions and the other reviewers. As a result, I keep my score unchanged. If accepted, I would hope that the authors add the numerical experiments discussed in the rebuttal (even if they are in the appendix) as well as add the references to the related work. %%%%%%%%%%%%%%%%%%%%%%%%%%%%%%%%% The goal of this paper is to use the proximal point method to solve an optimization problem of the form: min_x phi(x) s.t. g(x) \le eta where phi is mu-weakly convex and g is possibly a nonconvex function of the form g(x) = lambda ||x||_1 -h(x) (here h is convex). This formulation encodes minimize nonconvex function subject to sparsity constraints. The algorithm proposed by the authors is the proximal point algorithm. The authors provide an asymptotic convergence rate for the proposed algorithm.

Strengths: The idea of using a proximal point algorithm to solve constrained optimization problems is novel idea, but not groundbreaking given the recent work in proximal point methods for nonconvex functions. The claims the authors make seem grounded in theory. Although I did not go through the proofs in detail, I am confident that the results hold given my knowledge of proximal point methods.

Weaknesses: The approach of using a proximal method seems very similar to the work using Catalyst [Lin et al., Paquette et al.] in the nonconvex setting. Particularly, I am wondering about the following: (1). I know work which has used Catalyst (proximal point method) with Frank-Wolfe to accelerated Frank-Wolfe. How does this compare with doing Catalyst [Lin et al] say with the Frank-Wolfe method? (2). One major difficulty of working with mu-weakly convex functions is determining the mu, which is needed in the proximal point algorithm for the algorithm to be practical. In Catalyst paper with [Paquette, et al.], they addressed this difficulty. Do the author have any insight into how to find this mu? It would make the algorithm significantly more practical. (3). A major weakness in my opinion is that this is simply just the proximal point method applied to a constrained nonconvex proximal. I don't believe that the proof techniques are very different than the original proximal point method applied to weakly convex functions. In this regard, the proposed algorithm doesn't seem to make significant impart. Maybe the authors could address the importance of (1) their algorithm or (2) the techniques used?

Correctness: To the best of my knowledge, the theoretical and empirical claims made by the authors appear to be correct. Here are a few of my concerns: Confused with Table 1: Which terms are you exactly considering are smooth and convex? Is it g(x) or psi(x)? Also what are you measuring optimality with respect to? For solving purely non convex objective, typical measure of optimality is nearly stationary point. Assumption 2.1: I would add that this is the definition of a weakly convex function and point to references on the topic see Catalyst [Paquette et al] and references therein.

Clarity: The paper is well-written with enough details in the 8 pages for someone in the field to understand. There are a few typos (e.g. Line 141), but otherwise good.

Relation to Prior Work: The prior work is discussed in great detail. I would add the work done on the proximal point method for nonconvex functions Catalyst [Paquette et al] For your reference on subdifferentials I would add the book "Variational Analysis" by Rockafellar.

Reproducibility: Yes

Additional Feedback:


Review 4

Summary and Contributions: The authors study a new model with continuous nonconvex sparsity-inducing constraints. They present a proximal point algorithm that solves a sequence of convex subproblems with gradually relaxed constraint levels. The proposed algorithm achieves an O(1/\epsilon) (O(1/\epsilon^2)) complexity for deterministic (stochastic) optimization.

Strengths: The strengths of the paper are as follows. 1. The authors study a new model with continuous nonconvex sparsity-inducing constraints (see the problem (5) of this paper). The new model can explicitly impose the sparsity of solutions and is different from problems studied by previous works. 2. This paper is technically sound. The authors show the convergence and the complexity of the proposed method. 3. Experiments show the effectiveness of the proposed method.

Weaknesses: It is unclear why we need to define $g_k(x)$ as in Algorithm 1. The authors may want to clarify the motivation behind it. All $g(x)$ in the experiments are MCP penalty. The authors may want to conduct more experiments with other penalties, such as SCAD and Exp in Table 2. Though the paper is well written overall, some sentences and formulations are confusing. 1. In the abstract, the sentence “We also establish new convergence complexities to achieve an approximate KKT solution when the objective can be smooth/nonsmooth, deterministic/stochastic and convex/nonconvex with complexity that is on a par with gradient descent when applied to nonconvex regularized problem” is confusing. 2. The authors may want to change the notation $\psi’(x)$ in the inequality (6) to $\partial \psi(x), as they have assumed that $\psi(x)$ can be nonsmooth.

Correctness: The claims, the method, and the empirical methodology are correct.

Clarity: Overall, the paper is well written, while some sentences and formulations are confusing. Please refer to the Weakness part for more details.

Relation to Prior Work: The authors discuss the differences between this work and previous works in detail.

Reproducibility: Yes

Additional Feedback: ======Update After Rebuttal====== I have read the rebuttal and other reviewers' comments. The authors have addressed my concerns about experiments. They promise to compare with other nonconvex penalties in a later version. I will stick to my score.

[Author Response · NeurIPS 2020]

We thank reviewers for useful comments. **All reviewers:** We look at related questions: **(i) How is LCPP different**
**from proximal point and what role is played by $g_k$, (ii) Other experiments beyond MCP and on inverse problem**.
**(i)** LCPP is different from proximal point as it uses proximal point in the objective and convexification in constraint.
Since, we convexify the constraint in $g_k$, we can show that subproblem can be solved efficiently and approximate final
solution will be feasible. Feasibility allows to characterize a simple MFCQ condition under which $\bar{y}^k$ is bounded.
As demonstrated in Fig 1 below, (a)-(c) shows the original nonconvex set and convex subsets dynamically generated
in the run of algorithm. It is unclear whether these subsets will be algorithmically well behaved, i.e., the optimal
Lagrange multiplier for such constraints will be small. Indeed, as shown in Fig 1(d)-(e), as prox center gets close to
the point which violates MFCQ, the convex polyhedron flattens and bound on Lagrange multiplier blows up. Our
analysis precisely characterizes such bad conditions and can also provide a priori bounds, giving us full control over
complications that may arise in dynamically generated convex subproblems (see Appendix B). This need of bounded
$\bar{y}^k$ is crucial for convergence to KKT of constrained optimization and is not a concern for unconstrained or simple
set constrained problems in which usual proximal point operates. **(ii)** Due to the space limit in the submitted version,
we only report preliminary results on MCP with logistic regression objective. We will compare with other nonconvex
penalties and inverse problem objective in a later version.

Figure 1: (a)-(e) SCAD constrained optimization. Prox center denoted by green.

**Reviewer 1: Weakneess** The SCAD paper [12] showed the advantage of nonconvex surrogate penalty, and pointed out
the potential problem of biased estimation of large coefficients in Lasso since $\ell_1$ penalty does not flatten. Empirical
merit of using nonconvex constraint to relax $\ell_0$ is also demonstrated by examples in [25]. By suboptimality, we meant
to say that using nonconvex formulation is advantageous over $\ell_1$ due to the above discussions in the literature and used
it to merely motivate the main topic of this paper. We can always improve the language as pointed out by the reviewer.
Figure 3 does show that LCPP outperforms Lasso across a wide range of nonzero patterns. See below in other comment.
We agree that disclosing form of the constraint beforehand will make presentation better, succinct and allow to have
fairer comparison with [5,20] in Section 1. At first glance, it may seem that constraint form is too restrictive, however,
it covers variety of nonconvex sparsity inducing functions considered in the literature, e.g., [16,29]. They assume
$g(x) = g_1(x) - h(x)$ and we identify more structure that can be exploited to get efficient method for different problems.
**Related work** Thanks for pointing out 'CLASH' paper which we will add in the related work. While there exists more
advanced models for specific type of problems such as CLASH, they often require strong assumption on data (i.e. RIP)
and specific function type. In comparison, LCPP aims to deal with general loss and a variety of nonconvex penalties.
The proof technique and convergence results are substantially different from CLASH.
**Other Comment: Fig 3:** We observe that it is better to impose sparsity constraint. In gisette rcv1.binary and real-sim,
using too many features results in overfitting. In mnist, it is possible to obtain nearly the best performance with only
using nearly half the features. **Why (5) is hard:** Hardness is not merely in (5) but the question that we want a feasible
solution to (5) with faster convergence and with less restrictive constraint qualification that works for nonconvex
constraints of interest. Variety of algorithms in classic nonlinear programming [2] showed asymptotic convergence with
or without feasibility and [5,20] addressed the complexity part partially but the constraint qualifications requirements
were strong to ensure feasibility. We believe addressing these questions is hard for general problems but rates can be
further improved for particular class of problems. **Why choose (5) over (4):** While (5) controls the target penalty level
directly, (4) controls the level implicitly by tuning weight $\lambda$. Unlike the convex case, first-order optimal point of (4)
does not guarantee a small value of penalty for large $\lambda$. **Matrix rank:** Problem of rank constraint on matrices instead
of $\ell_0$-norm of vector is a natural extension. See [36] for more information. Very interesting comment.
**Reviewer 2:** W1. We compare with Logistic regression which uses liblinear (dual coordinate descent) through sklearn
interface. We also compare with GIST, an efficient nonconvex proximal gradient method for DC penalized problem.
These are quite standard solvers for convex and nonconvex sparse models. W2. In our problem, $\eta$ is a fixed target
constraint level and we don't change its value during the algorithm. By adding a constant to $\eta$, we are dealing with a
new problem with a more relaxed constraint. If same constant is added on both side of (5) then algorithm does not
change at all. W3. To the best of our knowledge, operator splitting and nonconvex ADMM methods are provably
convergent only when the constraint function is linear. They are not applicable for (5).
**Reviewer 3** Thank you for pointing out Catalyst. As per our knowledge, Frank-Wolfe (FW) method has complexity
guarantee if the constraint set is convex which isn't the case in (5). It seems possible to apply FW to solve the convex
subproblem in LCPP. We believe that adaptive strategy of Catalyst can be applied in our paper, since the subproblem is
convex and admits efficient proximal map. Table 1 describes the complexity to obtain an $\epsilon$-KKT points for different
types of objective function $\psi(x)$. The constraint $g(x)$ is nonconvex and nonsmooth according to Assumption 2.1.
**Reviewer 4:** We will change the language in the abstract. Other concerns addressed in common response.

[Meta-Review · NeurIPS 2020]

Dear authors, Thank you for submitting your paper. When producing the final camera-ready paper, please check the reviewer's remarks to make it stronger. Also, in the rebuttal you have committed to make the adjustments, so please do not forget about them :) Thank you